

# Jacobi–Lie $T$-plurality

**Jose J. Fernández-Melgarejo[1] ⋆ and Yuho Sakatani[2] †**

**1** Departamento de Física, Universidad de Murcia,
Campus de Espinardo, E-30100 Murcia, Spain
**2** Department of Physics, Kyoto Prefectural University of Medicine,
1-5 Shimogamohangi-cho, Sakyo-ku, Kyoto, Japan

⋆ melgarejo@um.es, † yuho@koto.kpu-m.ac.jp

## Abstract

We propose a Leibniz algebra, to be called DD$^+$, which is a generalization of the Drinfel'd double. We find that there is a one-to-one correspondence between a DD$^+$ and a Jacobi–Lie bialgebra, extending the known correspondence between a Lie bialgebra and a Drinfel'd double. We then construct generalized frame fields $E_A{}^M \in \mathrm{O}(D,D) \times \mathbb{R}^+$ satisfying the algebra $\hat{\mathcal{L}}_{E_A} E_B = -X_{AB}{}^C E_C$, where $X_{AB}{}^C$ are the structure constants of the DD$^+$ and $\hat{\mathcal{L}}$ is the generalized Lie derivative in double field theory. Using the generalized frame fields, we propose the Jacobi–Lie $T$-plurality and show that it is a symmetry of double field theory. We present several examples of the Jacobi–Lie $T$-plurality with or without Ramond–Ramond fields and the spectator fields.

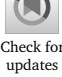

## 1   Introduction

Recently the Poisson–Lie $T$-duality [1, 2] or $T$-plurality [3] and their $U$-duality extensions [4–11] have been studied and developed by using the duality-covariant formulations, such as double field theory (DFT) [12–15] and its $U$-duality extensions. The Poisson–Lie $T$-duality is based on a Lie algebra called the Drinfel'd double while the $U$-duality variant is based on the exceptional Drinfel'd algebra (EDA) [4–7, 9, 16], which is an extension of the Drinfel'd double. Unlike the Drinfel'd double, the structure constants $X_{AB}{}^C$ of EDA do not necessarily have the antisymmetry, $X_{AB}{}^C \neq -X_{BA}{}^C$, and it is a Leibniz algebra rather than a Lie algebra. In this paper, we study a minimal extension of the Drinfel'd double by allowing the structure constants to admit the symmetric part $X_{(AB)}{}^C \neq 0$. Using this new Leibniz algebra, we study an extension of the Poisson–Lie $T$-duality, which we call the Jacobi–Lie $T$-plurality.[1]

The proposed Leibniz algebra has the form

$$
\begin{aligned}
T_a \circ T_b &= f_{ab}{}^c \, T_c \,, \qquad T^a \circ T^b = f_c{}^{ab} \, T^c \,, \\
T_a \circ T^b &= \left( f_a{}^{bc} + 2\,\delta_a^b \, Z^c - 2\,\delta_a^c \, Z^b \right) T_c - f_{ac}{}^b \, T^c + 2 Z_a \, T^b \,, \\
T^a \circ T_b &= -f_b{}^{ac} \, T_c + 2 Z^a \, T_b + \left( f_{bc}{}^a + 2\,\delta_b^a Z_c - 2\,\delta_c^a Z_b \right) T^c \,,
\end{aligned}
\tag{1.1}
$$

where $a = 1, \dots, D$, and $f_{ab}{}^c$ ($= -f_{ba}{}^c$) and $f_c{}^{ab}$ ($= -f_c{}^{ba}$) are the structure constants of two Lie subalgebras $\mathfrak{g}$ (generated by $T_a$) and $\tilde{\mathfrak{g}}$ (generated by $T^a$), respectively. This Leibniz algebra admits a symmetric bilinear form

$$
\langle T_a, T^b \rangle = \delta_a^b \,, \qquad \langle T_a, T_b \rangle = \langle T^a, T^b \rangle = 0 \,,
\tag{1.2}
$$

and two subalgebras $\mathfrak{g}$ and $\tilde{\mathfrak{g}}$ are maximally isotropic with respect to this. If $Z_a = Z^a = 0$, this algebra reduces to the Lie algebra of the Drinfel'd double, but otherwise the "adjoint-invariance" is relaxed as follows by allowing for a scale transformation:

$$
\delta_A \langle T_B, T_C \rangle \equiv \langle T_A \circ T_B, T_C \rangle + \langle T_B, T_A \circ T_C \rangle = 2 Z_A \langle T_B, T_C \rangle \,,
\tag{1.3}
$$

where $T_A \equiv (T_a, T^a)$ ($A = 1, \dots, 2D$) and $Z_A \equiv (Z_a, Z^a)$. Since this Leibniz algebra is an extension of the Drinfel'd double by admitting the scale symmetry $\mathbb{R}^+$, we call this an extended

---

[1]The Jacobi–Lie $T$-duality studied in [17, 18] is very similar to our proposal, and this paper is strongly inspired by these papers. However, our identification of the supergravity fields is different from the one given in [17, 18]. The details are explained in sections 3 and 4.

Drinfel'd algebra $DD^+$. It turns out that this $\mathbb{R}^+$ symmetry provides a scale factor similar to the trombone symmetry in supergravity [19].

In this paper, we show that the $DD^+$ provides an alternative way to define the Jacobi–Lie algebra, and explain how to construct geometric objects such as the Jacobi–Lie structures from a given $DD^+$. We also show that we can systematically construct the generalized frame fields $E_A{}^M$ satisfying the frame algebra

$$\hat{\mathcal{L}}_{E_A} E_B = -X_{AB}{}^C E_C \,, \tag{1.4}$$

where $\hat{\mathcal{L}}$ denotes the generalized Lie derivative in DFT and $X_{AB}{}^C$ are the structure constants of the $DD^+$. Similar to the recent studies on the Poisson–Lie $T$-duality/$T$-plurality in the context of DFT [20–22], exploiting the relation (1.4), we show that the Jacobi–Lie $T$-plurality is a symmetry of type II DFT.

At the level of the supergravity (or more precisely, DFT), the proposed Jacobi–Lie $T$-duality is indeed a symmetry of the equations of motion even if the Ramond–Ramond (R–R) fields or spectator fields are present. However, at the level of the string sigma model, due to the presence of the scale factor, we find difficulty in showing the covariance of the equations of motion under the Jacobi–Lie $T$-plurality. We discuss this issue from several approaches and also discuss the relation to the Jacobi–Lie $T$-duality proposed in [17].

This paper is organized as follows. In section 2, after introducing the Leibniz algebra $DD^+$, we explain how to construct the Jacobi–Lie structures and the generalized frame fields from the $DD^+$. We find that the generalized frame fields $E_A{}^M$ have a dependence on the dual coordinates $\tilde{x}_m$ of the doubled space (although the section condition of DFT is not broken). We also consider several examples of $DD^+$ and explicitly construct the Jacobi–Lie structures and the generalized frame fields $E_A{}^M$. A relation between the $DD^+$ and embedding tensors in gauged supergravities is also briefly discussed. In section 3, we provide a definition of the Jacobi–Lie symmetric backgrounds and show that the equations of motion of DFT have a manifest symmetry under the Jacobi–Lie $T$-plurality. For convenience, we provide several concrete examples of the Jacobi–Lie $T$-plurality with and without the R–R fields or the spectator fields. In section 4, we discuss the issue of the Jacobi–Lie $T$-plurality in the string sigma model. Section 5 is devoted to conclusion and discussion.

## 2 Jacobi–Lie structures

In this section, we propose a Leibniz algebra $DD^+$ and construct several quantities, such as the Jacobi–Lie structure, which play an important role in the Jacobi–Lie $T$-plurality. In section 2.3, we clarify the relation between the $DD^+$ and the Jacobi–Lie bialgebra studied in [23–26]. Several examples are given in section 2.4. In section 2.5, we comment on a relation between $DD^+$ and embedding tensors in half-maximal 7D gauged supergravity.

### 2.1 Algebra

A (classical) Drinfel'd double can be defined as a $2D$-dimensional Lie algebra $\mathfrak{d}$ which admits an adjoint-invariant metric $\langle \cdot, \cdot \rangle$ and allows a decomposition $\mathfrak{d} = \mathfrak{g} \oplus \tilde{\mathfrak{g}}$, where $\mathfrak{g}$ and $\tilde{\mathfrak{g}}$ form Lie subalgebras that are maximally isotropic with respect to $\langle \cdot, \cdot \rangle$. We choose the basis $T_a \in \mathfrak{g}$ and $T^a \in \tilde{\mathfrak{g}}$ such that the metric becomes $\langle T_a, T^b \rangle = \delta_a^b$, and denote the subalgebras as $[T_a, T_b] = f_{ab}{}^c T_c$ and $[T^a, T^b] = f_c{}^{ab} T^c$. Then, from the adjoint invariance

$$\langle [T_A, T_B], T_C \rangle + \langle T_B, [T_A, T_C] \rangle = 0 \,, \tag{2.1}$$

we can determine the mixed-commutator as

$$[T_a, T^b] = f_a{}^{bc} T_c - f_{ac}{}^b T^c. \tag{2.2}$$

The adjoint-invariant metric can be expressed as

$$\langle T_A, T_B \rangle = \eta_{AB}, \qquad \eta_{AB} = \begin{pmatrix} 0 & \delta_a^b \\ \delta_b^a & 0 \end{pmatrix}, \tag{2.3}$$

and we raise or lower the indices $A, B$ by using $\eta_{AB}$ and its inverse $\eta^{AB}$.

Now, let us introduce the Leibniz algebra DD$^+$,

$$T_A \circ T_B = X_{AB}{}^C T_C. \tag{2.4}$$

We keep assuming that $\mathfrak{g}$ and $\tilde{\mathfrak{g}}$ are maximally-isotropic Lie subalgebras but relax the adjoint-invariance as in Eq. (1.3). We then find that the structure constants should have the form

$$X_{AB}{}^C \equiv F_{AB}{}^C + Z_A \delta_B^C - Z_B \delta_A^C + \eta_{AB} Z^C, \tag{2.5}$$

where $F_{AB}{}^C = F_{ABD} \eta^{DC}$, $F_{ABC} = F_{[ABC]}$, and $F_{ABC}$ has the only non-vanishing components $F_{ab}{}^c$ and $F_a{}^{bc}$. Defining $f_{ab}{}^c$ and $f_c{}^{ab}$ through $T_a \circ T_b = f_{ab}{}^c T_c$ and $T^a \circ T^b = f_c{}^{ab} T^c$, we can parameterize $F_{ABC}$ as

$$F_{abc} = 0, \quad F_{ab}{}^c = f_{ab}{}^c - Z_a \delta_b^c + Z_b \delta_a^c, \quad F_a{}^{bc} = f_a{}^{bc} - \delta_a^b Z^c + Z^b \delta_a^c, \quad F^{abc} = 0, \tag{2.6}$$

where $Z_A = (Z_a, Z^a)$. By substituting these into Eq. (2.4), we obtain the algebra (1.1).

The closure conditions, or the Leibniz identities,

$$T_A \circ (T_B \circ T_C) = (T_A \circ T_B) \circ T_C + T_B \circ (T_A \circ T_C), \tag{2.7}$$

require the following identities for the structure constants:

$$f_{[ab}{}^e f_{c]e}{}^d = 0, \qquad f_e{}^{[ab} f_d{}^{c]e} = 0, \tag{2.8}$$

$$4 f_{[a}{}^{e[c} f_{b]e}{}^{d]} - f_{ab}{}^e f_e{}^{cd} + 4 f_{[a}{}^{cd} Z_{b]} + 4 f_{ab}{}^{[c} Z^{d]} + 8 f_{e[a}{}^{[c} \delta_{b]}^{d]} Z^e - 16 Z_{[a} \delta_{b]}^{[c} Z^{d]} = 0, \tag{2.9}$$

$$f_{ab}{}^c Z_c = 0, \qquad f_a{}^{bc} Z_c = f_{ac}{}^b Z^c, \qquad Z^c f_c{}^{ab} = 0, \qquad Z^a Z_a = 0. \tag{2.10}$$

## 2.2 Generalized frame fields

Here we construct the generalized frame fields $E_A{}^M$. We introduce a group element $g = e^{x^a T_a}$ and define the left-/right-invariant 1-forms as

$$\ell = \ell_m^a \, dx^m \, T_a = g^{-1} dg, \qquad r = r_m^a \, dx^m \, T_a = dg \, g^{-1}. \tag{2.11}$$

Their inverse matrices are denoted as $v_a^m$ and $e_a^m$ ($v_a^m \ell_m^b = \delta_a^b = e_a^m r_m^b$). We then consider the adjoint-like action as

$$g \triangleright T_A \equiv e^{x^b T_b \circ} T_A = T_A + x^b T_b \circ T_A + \frac{1}{2!} x^b T_b \circ (x^c T_c \circ T_A) + \cdots, \tag{2.12}$$

and define

$$g^{-1} \triangleright T_A \equiv M_A{}^B(g) T_B. \tag{2.13}$$

It turns out that this matrix $M_A{}^B$ can be parameterized as

$$M_A{}^B \equiv \begin{pmatrix} a_a{}^b & 0 \\ -\pi^{ac} a_c{}^b & e^{-2\Delta}(a^{-1})_b{}^a \end{pmatrix}, \tag{2.14}$$

where $\pi^{ab}$ is an antisymmetric field: $\pi^{ab} = -\pi^{ba}$.

Similar to the case of the Drinfel'd double [27] (see also [9] for a general discussion), we find that $a_a{}^b$, $\pi^{ab}$, and $\Delta$ satisfy the algebraic identities

$$f_{ab}{}^c = a_a{}^d\, a_b{}^e\, (a^{-1})_f{}^c\, f_{de}{}^f\,, \tag{2.15}$$

$$f_d{}^{[ab}\,\pi^{c]d} + f_{de}{}^{[a}\,\pi^{b|d|}\,\pi^{c]e} - 2\,\pi^{[ab}\,\pi^{c]d}\,Z_d + 2\,\pi^{[ab}\,Z^{c]} = 0\,, \tag{2.16}$$

$$f_a{}^{bc} = \mathrm{e}^{-2\Delta}\, a_a{}^d\, (a^{-1})_e{}^b\, (a^{-1})_f{}^c\, f_d{}^{ef} + 2\,f_{ad}{}^{[b}\,\pi^{c]d} + 6\,\delta_a^{[b}\,\pi^{cd]}\,Z_d\,, \tag{2.17}$$

$$a_a{}^b\, Z_b = Z_a\,, \qquad Z^a + \pi^{ab}\, Z_b = \mathrm{e}^{-2\Delta}(a^{-1})_b{}^a\, Z^b \qquad \left(\Leftrightarrow M_A{}^B\, Z_B = Z_A\right)\,, \tag{2.18}$$

and the differential identities

$$D_a\Delta = Z_a\,, \qquad D_a a_b{}^c = -f_{ab}{}^d\, a_d{}^c\,, \tag{2.19}$$

$$D_a\pi^{bc} = f_a{}^{bc} + 2\,f_{ad}{}^{[b}\,\pi^{|d|c]} - 2\,Z_a\,\pi^{bc} - 4\,Z^{[b}\,\delta_a^{c]}\,, \tag{2.20}$$

where $D_a \equiv e_a^m\,\partial_m$. Combining these identities, we also find

$$\pounds_{v_a}\Delta = Z_a\,, \qquad \pounds_{v_a} a_b{}^c = -a_b{}^d\, f_{ad}{}^c\,, \tag{2.21}$$

$$\pounds_{v_a}\pi^{mn} = \left(f_a{}^{bc} + 2\,\delta_a^b\, Z^c - 2\,\delta_a^c\, Z^b\right) v_b^m\, v_c^n + 2\,Z_a\,\pi^{mn}\,. \tag{2.22}$$

Here we have defined

$$\pi^{mn} \equiv \mathrm{e}^{2\Delta}\,\pi^{ab}\, e_a^m\, e_b^n\,, \tag{2.23}$$

which turns out to be a Jacobi–Lie structure.

Now we define the generalized frame fields as

$$E_A{}^M \equiv M_A{}^B\, V_B{}^M\,, \qquad V_A{}^M \equiv \begin{pmatrix} v_a^m & 0 \\ 0 & \ell_m^a \end{pmatrix}, \tag{2.24}$$

and obtain

$$E_A{}^M = \begin{pmatrix} e_a^m & 0 \\ -\pi^{ab}\, e_b^m & \mathrm{e}^{-2\Delta}\, r_m^a \end{pmatrix}. \tag{2.25}$$

If $Z^a = 0$, these generalized frame fields satisfy the relation

$$\hat{\pounds}_{E_A} E_B{}^M = -X_{AB}{}^C\, E_C{}^M\,, \tag{2.26}$$

by means of the generalized Lie derivative in DFT,

$$\hat{\pounds}_V W^M \equiv V^N\,\partial_N W^M - (\partial_N V^M - \partial^M V_N)\, W^N\,, \tag{2.27}$$

where $\partial_M \equiv (\partial_m, \tilde{\partial}^m)$ are partial derivatives with respect to the doubled coordinates $x^M \equiv (x^m, \tilde{x}_m)$ and the indices $M, N$ are raised or lowered with the metric $\eta_{MN}$ (which is the same matrix as $\eta_{AB}$). In the presence of $Z^a$, we need to modify the generalized frame fields as

$$E_A{}^M \equiv \begin{pmatrix} e_a^m & 0 \\ -\pi^{ac}\, e_c^m & \mathrm{e}^{-2\omega}\, r_m^a \end{pmatrix}, \qquad \mathrm{e}^{-2\omega} \equiv \mathrm{e}^{-2\Delta}\,\tilde{\sigma}\,, \tag{2.28}$$

where $\tilde{\sigma}$ is supposed to be positive. If this $\tilde{\sigma}$ satisfies

$$\partial_m\tilde{\sigma} = 0\,, \qquad \tilde{\partial}^m\tilde{\sigma} \equiv -2\,Z^m \equiv -2\,Z^a\, v_a^m\,, \tag{2.29}$$

we find that the new generalized frame fields satisfy the desired relation (2.26).

Since the modified generalized frame fields depend on the dual coordinates $\tilde{x}_m$, one may be concerned about the section condition (i.e., a consistency condition in DFT). However, we can easily show that the section condition is not broken. As we discuss later, the supergravity fields are constructed from $E_A{}^M$ which is composed of the fields $\{\Delta, \tilde{\sigma}, e_a^m, \pi^{mn}\}$.[2] Using $£_Z = Z^a \, £_{v_a}$, the differential identities, and the Leibniz identities, we find

$$£_Z \Delta = Z^a \, Z_a = 0, \quad £_Z e_a^m = Z^b \, £_{v_b} e_a^m = 0,$$
$$£_Z \pi^{mn} = Z^a \left(f_a{}^{bc} + 2\,\delta_a^b \, Z^c - 2\,\delta_a^c \, Z^b\right) v_b^m \, v_c^n + 2\, Z^a \, Z_a \, \pi^{mn} = 0. \tag{2.30}$$

Therefore, $Z$ is a Killing vector field and we can choose the coordinate system such that $Z = \partial_w$ is realized. Then all of the fields $\phi$ are independent of the coordinate $w$. In this coordinate system, we can explicitly find $\tilde{\sigma} = -2\,\tilde{w} + \text{const.}$, and then the section condition reduces to

$$0 = \eta^{MN} \, \partial_M \tilde{\sigma} \, \partial_N \phi = -2\, \partial_w \phi. \tag{2.31}$$

This is indeed satisfied because fields $\phi$ are independent of $w$ due to the Killing equation.

Let us also show several properties of the bi-vector field $\pi \equiv \frac{1}{2}\, \pi^{mn} \, \partial_m \wedge \partial_n$. By using the differential and algebraic identities, we can show

$$[\pi, \pi]_S = 2\, E \wedge \pi, \qquad [E, \pi]_S = 0, \tag{2.32}$$

where $E \equiv -2\, Z^a \, e_a$ and we have defined the Schouten–Nijenhuis bracket for a $p$-vector $v$ and a $q$-vector $w$ as

$$[v, w]_S^{m_1 \cdots m_{p+q-1}} \equiv \frac{(p+q-1)!}{(p-1)!\, q!} \, v^{p[m_1 \cdots m_{p-1}} \, \partial_p w^{m_p \cdots m_{p+q-1}]}$$
$$+ \frac{(-1)^{pq}(p+q-1)!}{(q-1)!\, p!} \, w^{p[m_1 \cdots m_{q-1}} \, \partial_p v^{m_q \cdots m_{p+q-1}]}, \tag{2.33}$$

or more explicitly,

$$[\pi, \pi]_S \equiv \pi^{q[m} \, \partial_q \pi^{np]} \, \partial_m \wedge \partial_n \wedge \partial_p, \qquad [E, \pi]_S \equiv \frac{1}{2!} \, £_E \pi^{mn} \, \partial_m \wedge \partial_n. \tag{2.34}$$

The first property is equivalent to the absence of the non-geometric $R$-flux

$$X^{abc} = 3\, \pi^{d[a} \, D_d \pi^{bc]} + 3\, f_{de}{}^{[a} \, \pi^{b|d|} \, \pi^{c]e} - 6\, \pi^{[ab} \, \pi^{c]d} \, D_d \Delta - 3\, a_d{}^{[a} \, \pi^{bc]} \, r_m^d \, \tilde{\partial}^m \tilde{\sigma} = 0, \tag{2.35}$$

and the second one follows from

$$£_{e_a} \pi^{mn} = e^{2\Delta} \left(f_a{}^{bc} - 4\, Z^{[b} \, \delta_a^{c]}\right) e_b^m \, e_c^n. \tag{2.36}$$

If we define a bracket

$$\{f, g\} \equiv \pi^{mn} \, \partial_m f \, \partial_n g + f \, E^m \, \partial_m g - g \, E^m \, \partial_m f, \tag{2.37}$$

for any functions $f$ and $g$, the Jacobi identity

$$\{f, \{g, h\}\} + \{g, \{h, f\}\} + \{h, \{f, g\}\} = 0, \tag{2.38}$$

becomes

$$\left(3\, \pi^{[m|q|} \, \partial_q \pi^{np]} + 3\, E^{[m} \, \pi^{np]}\right) \partial_m f \, \partial_n g \, \partial_p h$$
$$+ \left(f \, \partial_m g \, \partial_n h + g \, \partial_m h \, \partial_n f + h \, \partial_m f \, \partial_n g\right) £_E \pi^{mn} = 0. \tag{2.39}$$

---

[2] In the presence of the dilaton and the Ramond–Ramond fields, there are additional fields which should be chosen such that the section condition is not broken.

In particular, by choosing a constant function $f = \text{const.}$, the Jacobi identity requires $\pounds_E \pi^{mn} = 0$, and then the Jacobi identity is equivalent to the conditions (2.32). The bracket (2.37) is known as the Jacobi bracket and accordingly, the pair of the bi-vector field $\pi^{mn}$ and the vector field $E$ satisfying Eq. (2.32) is called the Jacobi structure. In particular, when $E = 0$, the Jacobi bracket/structure reduces to the usual Poisson bracket/structure. To consistently define the Jacobi structure on a group manifold $G = \exp \mathfrak{g}$, properties (2.22), called the multiplicativity [24], need to be satisfied. In our construction, the multiplicativity is automatically satisfied, and then this kind of Jacobi structure is called the Jacobi–Lie structure.

As it has been studied in [23,24,26], the Leibniz identity (2.9) can be regarded as a cocycle condition, and it is automatically satisfied if we consider the coboundary ansatz

$$f_a{}^{bc} = 2\, r^{[b|d|} f_{ad}{}^{c]} - 2 Z_a\, r^{bc} + 4 Z^{[b}\, \delta_a^{c]},\qquad(2.40)$$

where $r^{ab}$ is a skew-symmetric constant matrix. The other Leibniz identities (under $f_{[ab}{}^e f_{c]e}{}^d = 0$ and $f_{ab}{}^c Z_c = 0$) are equivalent to[3]

$$r^{ab} Z_b = Z^a,\quad Z^c f_{cd}{}^{[a} r^{b]d} = 0,\quad \text{CYBE}^{abc} \equiv 3 f_{de}{}^{[a} r^{b|d|} r^{c]e} - 6 Z^{[a} r^{bc]} = 0,\qquad(2.41)$$

which are known as the generalized classical Yang–Baxter equations [24]. For this type of algebra, we can find the solution of the differential equation (2.22) as

$$\pi^{mn} = r^{ab} \left( v_a^m v_b^n - \mathrm{e}^{2\Delta} e_a^m e_b^n \right).\qquad(2.42)$$

We note that this type of Jacobi–Lie structures (associated with coboundary-type algebras) has been studied in [24] (see also [17,26]).

## 2.3 Jacobi–Lie bialgebra

Let us explain the relation between $DD^+$ and the Jacobi–Lie bialgebra studied in [23–26]. We begin with a Lie algebra $\mathfrak{g}$ with commutation relation $[T_a, T_b] = f_{ab}{}^c T_c$. We introduce the dual space $\mathfrak{g}^*$ spanned by $\{T^a\}$ and suppose that they form a Lie algebra $[T^a, T^b] = f_c{}^{ab} T^c$. We introduce the differentials $\mathrm{d}$ and $\mathrm{d}_*$ which acts on $\mathfrak{g}^*$ and $\mathfrak{g}$ as

$$\mathrm{d}T^a = -\tfrac{1}{2} f_{bc}{}^a\, T^b \wedge T^c,\qquad \mathrm{d}_* T_a = -\tfrac{1}{2} f_a{}^{bc}\, T_b \wedge T_c,\qquad(2.43)$$

and 1-cocycles $X_0 \in \mathfrak{g}$ and $\phi_0 \in \mathfrak{g}^*$ satisfying $\mathrm{d}_* X_0 = 0$ and $\mathrm{d}\phi_0 = 0$. We then define

$$\mathrm{d}_{*X_0} \equiv \mathrm{d}_* + X_0 \wedge,\qquad(2.44)$$

and a bracket $[\cdot,\cdot]_{\phi_0}$ for $x \in \wedge^p \mathfrak{g}$ and $y \in \wedge^q \mathfrak{g}$ as

$$[x, y]_{\phi_0} = [x, y] + (-1)^{p-1}(p-1)\, x \wedge \iota_{\phi_0} y - (q-1)\, \iota_{\phi_0} x \wedge y,\qquad(2.45)$$

where $[\cdot,\cdot]$ is the algebraic Schouten bracket and $\iota_{\phi_0}$ denotes the contraction. Using these, we can define a Jacobi–Lie bialgebra as a pair $((\mathfrak{g}, \phi_0),(\mathfrak{g}^*, X_0))$ which satisfies

$$\begin{aligned}
\mathrm{d}_{*X_0}[x, y] &= [x, \mathrm{d}_{*X_0} y]_{\phi_0} - [y, \mathrm{d}_{*X_0} x]_{\phi_0},\\
\langle \phi_0, X_0 \rangle &= 0,\qquad \iota_{\phi_0}(\mathrm{d}_* x) + [X_0, x] = 0,
\end{aligned}\qquad(2.46)$$

for any elements $x, y \in \mathfrak{g}$. If we expand $X_0$ and $\phi_0$ as

$$X_0 = \alpha^a\, T_a,\qquad \phi_0 = \beta_a\, T^a,\qquad(2.47)$$

---

[3]The first equation is implied by $\left(f_{ac}{}^b - 2 Z_a \delta_c^b\right)\left(Z^c - r^{cd} Z_d\right) = 0$. The last equation can be relaxed as $f_{de}{}^{[a} \text{CYBE}^{|e|bc]} = 0$ if $Z_a = 0$. Indeed, in the case of six-dimensional Jacobi–Lie bialgebras [26], an algebra satisfying $\text{CYBE}^{abc} \neq 0$ (i.e., a quasitriangular coboundary Jacobi–Lie bialgebra) is realized only when $Z_a = 0$.

the 1-cocycle conditions $d_* X_0 = 0$ and $d\phi_0 = 0$ are equivalent to

$$\alpha^a f_a{}^{bc} = 0, \qquad \beta_a f_{bc}{}^a = 0, \qquad (2.48)$$

and the conditions (2.46) can be expressed as

$$4 f_{[a}{}^{e[c} f_{b]e}{}^{d]} - f_{ab}{}^e f_e{}^{cd} + 2 f_{[a}{}^{cd} \beta_{b]} + 2 f_{ab}{}^{[c} \alpha^{d]} + 4 f_{e[a}{}^{[c} \delta_{b]}^{d]} \alpha^e - 4 \beta_{[a} \delta_{b]}^{[c} \alpha^{d]} = 0,$$
$$\alpha^a \beta_a = 0, \qquad \alpha^c f_{ca}{}^b - \beta_c f_a{}^{cb} = 0. \qquad (2.49)$$

They are exactly the same as the Leibniz identities of the $\mathrm{DD}^+$ under the identification

$$\alpha^a = 2 Z^a, \qquad \beta_a = 2 Z_a. \qquad (2.50)$$

This shows that there is a one-to-one correspondence between a Leibniz algebra $\mathrm{DD}^+$ and a Jacobi–Lie bialgebra. In [25], by using a generalized Courant bracket, commutation relations

$$[T_a, T_b] = f_{ab}{}^c T_c, \qquad [T^a, T^b] = f_c{}^{ab} T^c,$$
$$[T_a, T^b] = \left(f_a{}^{bc} + \tfrac{1}{2} \alpha^c \delta_a^b - \alpha^b \delta_a^c\right) T_c + \left(f_{ca}{}^b - \tfrac{1}{2} \beta_c \delta_a^b + \beta_a \delta_c^b\right) T^c, \qquad (2.51)$$

are introduced, but in general, the Jacobi identities are not satisfied and this bracket does not define a Lie algebra. Rather, this can be regarded as the antisymmetric part of the Leibniz algebra $\mathrm{DD}^+$,

$$[T_A, T_B] \equiv \tfrac{1}{2} \left(T_A \circ T_B - T_B \circ T_A\right). \qquad (2.52)$$

As we discussed in section 2.2, a $\mathrm{DD}^+$ allows us to systematically construct the Jacobi–Lie structure $\pi^{mn}$ for a general Jacobi–Lie bialgebra. In [17], a similar construction has been attempted by using the commutation relations (2.51). However, due to the absence of the symmetric part $X_{(AB)}{}^C$ of the structure constants, it was not successful, and only the coboundary-type algebras have been studied, where $\pi^{mn}$ has the simple expression (2.42). A $\mathrm{DD}^+$ also allows us to obtain the scale factor $\Delta$ from a straightforward computation of the matrix $M_A{}^B$, and these are the advantage of our approach based on the Leibniz algebra. In the next subsection, as a demonstration, we explicitly compute the Jacobi–Lie structures for several concrete examples.

## 2.4 Examples of Jacobi–Lie structures

The low-dimensional Jacobi–Lie groups have been classified in [25], and in particular, classifications of the coboundary-type Jacobi–Lie groups have been given in [26]. For the coboundary-type algebras, there is a general formula (2.42) for the Jacobi–Lie structures, and here we consider two examples of Leibniz algebras that are not of the coboundary type.

**(I)** $((\mathrm{IV}, -\epsilon \tilde{X}^1), (\mathrm{IV.i}, -\epsilon \alpha X_3))$

Let us consider $((\mathrm{IV}, -\epsilon \tilde{X}^1), (\mathrm{IV.i}, -\epsilon \alpha X_3))$ $(\alpha > 0)$ in Table 6 of [25], which corresponds to

$$f_{12}{}^2 = -f_{12}{}^3 = f_{13}{}^3 = -1, \quad f_1{}^{13} = f_2{}^{23} = \alpha, \quad f_1{}^{23} = 1, \quad Z^3 = -\tfrac{\epsilon \alpha}{2}, \quad Z_1 = -\tfrac{\epsilon}{2}. \quad (2.53)$$

The Leibniz identities require $\epsilon = 1$ or $\epsilon = 2$. While $\epsilon = 1$ gives a coboundary algebra, here we consider the non-coboundary case $\epsilon = 2$.

Using $g = e^{x T_1} e^{y T_2} e^{z T_3}$, the left-/right-invariant vectors are found as

$$v_1 = \partial_x + y \partial_y + (z - y) \partial_z, \quad v_2 = \partial_y, \quad v_3 = \partial_z,$$
$$e_1 = \partial_x, \quad e_2 = e^x (\partial_y - x \partial_z), \quad e_3 = e^x \partial_z, \qquad (2.54)$$

and by computing the matrix $M_A{}^B$, we find

$$\pi = \left[\alpha\,(\mathrm{e}^{-x}-1)\,\partial_x + (x-\alpha\,y)\,\partial_y\right]\wedge\partial_z\,, \qquad \mathrm{e}^{-2\Delta} = \mathrm{e}^{2x}\,. \tag{2.55}$$

From $\tilde{\partial}^m\tilde{\sigma} = -2\,Z^a\,v_a^m$ we can easily find

$$\tilde{\sigma} = 2\,\alpha\,\tilde{z} + \mathrm{const.,} \tag{2.56}$$

and then we find that the generalized frame fields enjoy the algebra $\hat{\mathcal{L}}_{E_A}E_B = -X_{AB}{}^C E_C$.

**(II)** $((\mathrm{III}, -2\tilde{X}^1), (\mathrm{III.ii}, -(X_2+X_3)))$

Another example is $((\mathrm{III}, -2\tilde{X}^1), (\mathrm{III.ii}, -(X_2+X_3)))$ of [25], which corresponds to

$$f_{12}{}^2 = f_{12}{}^3 = f_{13}{}^2 = f_{13}{}^3 = -1\,, \quad f_1{}^{12} = f_1{}^{13} = 1\,, \quad Z^2 = Z^3 = -\tfrac{1}{2}\,, \quad Z_1 = -1\,. \tag{2.57}$$

Using $g = \mathrm{e}^{x\,T_1}\,\mathrm{e}^{y\,T_2}\,\mathrm{e}^{z\,T_3}$, the left-/right-invariant vectors are found as

$$\begin{aligned} v_1 &= \partial_x + (y+z)(\partial_y + \partial_z)\,, & v_2 &= \partial_y\,, & v_3 &= \partial_z\,, \\ e_1 &= \partial_x\,, & e_2 &= \mathrm{e}^x(\cosh x\,\partial_y + \sinh x\,\partial_z)\,, & e_3 &= \mathrm{e}^x(\sinh x\,\partial_y + \cosh x\,\partial_z)\,. \end{aligned} \tag{2.58}$$

From the matrix $M_A{}^B$ and $\tilde{\partial}^m\tilde{\sigma} = -2\,Z^a\,v_a^m$, we find

$$\pi = (z-y)\,\partial_y \wedge \partial_z\,, \qquad \mathrm{e}^{-2\Delta} = \mathrm{e}^{2x}\,, \qquad \tilde{\sigma} = \tilde{y} + \tilde{z} + \mathrm{const.,} \tag{2.59}$$

and then the generalized frame fields satisfy the algebra $\hat{\mathcal{L}}_{E_A}E_B = -X_{AB}{}^C E_C$.

In this way, for a given Leibniz algebra, we can easily compute the Jacobi–Lie structure and the generalized frame fields.

## 2.5 Embedding tensor in half-maximal 7D gauged supergravity

As a side remark, we here clarify the relation between six-dimensional DD$^+$s and the embedding tensors in half-maximal 7D gauged supergravity. In [28], embedding tensors in half-maximal 7D gauged supergravity have been classified, where the duality group is $\mathrm{O}(3,3)\times\mathbb{R}^+$. In our convention, their embedding tensor can be expressed as

$$\begin{aligned} X_{AB}{}^C &\equiv F_{AB}{}^C + Z_A\,\delta_B^C - Z_B\,\delta_A^C + \eta_{AB}\,Z^C\,, \\ F_{abc} &= H_{abc}\,, \quad F_{ab}{}^c = f_{ab}{}^c - Z_a\,\delta_b^c + Z_b\,\delta_a^c\,, \\ F_a{}^{bc} &= f_a{}^{bc} - \delta_a^b\,Z^c + Z^b\,\delta_a^c\,, \quad F^{abc} = R^{abc}\,, \end{aligned} \tag{2.60}$$

where the non-vanishing components are

$$\begin{aligned} H_{123} &= Q_{11}\,, & f_1{}^{23} &= Q_{22}\,, & f_2{}^{13} &= -Q_{33}\,, & f_3{}^{12} &= Q_{44}\,, & Z_1 &= -\xi_0\,, \\ R^{123} &= \tilde{Q}^{11}\,, & f_{23}{}^1 &= \tilde{Q}^{22}\,, & f_{13}{}^2 &= -\tilde{Q}^{33}\,, & f_{12}{}^3 &= \tilde{Q}^{44}\,, & f_{12}{}^2 &= f_{13}{}^3 = -\xi_0\,. \end{aligned} \tag{2.61}$$

The possible values of $Q_{ij}$, $\tilde{Q}^{ij}$, and $\xi_0$ have been classified in Table 2 of [28] and there are 13 inequivalent solutions, which are called orbits (see Appendix A).

Using Eq. (2.60). we can define a Leibniz algebra $T_A \circ T_B = X_{AB}{}^C T_C$ that admits the usual bilinear form $\langle T_A, T_B \rangle = \eta_{AB}$. Due to the presence of $F_{abc}$ and $F^{abc}$, this is not an algebra of a DD$^+$. However, as we explain in Appendix A, by performing an $\mathrm{O}(3,3)$ redefinition of the generators, most of the 13 orbits can be mapped to some DD$^+$s. As a demonstration, let us take orbit 10, where $Q_{ii}$ and $\tilde{Q}^{ii}$ are given by

$$\frac{Q_{ii}}{\cos\alpha} = (1,-1,0,0)\,, \qquad \frac{\tilde{Q}^{ii}}{\sin\alpha} = (0,0,1,-1) \qquad \left(-1 \le \xi_0 \le 1,\ -\tfrac{\pi}{4} < \alpha \le \tfrac{\pi}{4}\right). \tag{2.62}$$

Performing a redefinition of the generators $T_A \to C_A{}^B T_B$ with

$$C_A{}^B = \begin{pmatrix} -\frac{1}{\cos\alpha} & 0 & 0 & 0 & 0 & 0 \\ 0 & \frac{1}{\sqrt{2}} & -\frac{1}{\sqrt{2}} & 0 & 0 & 0 \\ 0 & 0 & 0 & 0 & \frac{1}{\sqrt{2}} & \frac{1}{\sqrt{2}} \\ 0 & 0 & 0 & -\cos\alpha & 0 & 0 \\ 0 & 0 & 0 & 0 & \frac{1}{\sqrt{2}} & -\frac{1}{\sqrt{2}} \\ 0 & \frac{1}{\sqrt{2}} & \frac{1}{\sqrt{2}} & 0 & 0 & 0 \end{pmatrix} \in O(3,3), \tag{2.63}$$

we find that the structure constants become

$$f_{12}{}^3 = -1, \quad f_{13}{}^2 = -1, \quad f_{12}{}^2 = f_{13}{}^3 = \frac{\xi_0 - \sin\alpha}{\cos\alpha}, \quad Z_1 = \frac{\xi_0}{\cos\alpha}. \tag{2.64}$$

For example, if $\xi_0 = \sin\alpha$ or $\frac{\xi_0 - \sin\alpha}{\cos\alpha} = -1$ is realized, this is equivalent to a Jacobi–Lie bialgebra $((VI_0, bX_3),(I,0))$ or $((III, bX_1),(I,0))$ given in Table 7 of [25], respectively. By choosing another matrix $C_A{}^B$, we may also find another Jacobi–Lie bialgebra classified in [25]. In this sense, the flux algebra given in Eqs. (2.60) and (2.61) can be mapped to a $DD^+$. Then, as we discussed in the previous section, we can systematically construct the generalized frame fields (or twist matrix) by using the Jacobi–Lie structure.

A similar analysis can be carried out for any (half-)maximal $d$-dimensional supergravities, because the $T$-duality-covariant flux $F_{ABC}$ is always contained in the embedding tensor and the role of $Z_A$ can be played by the trombone gauging [29–31] or the dilaton flux. In particular, the half-maximal $d = 6, 5, 4$ supergravities explicitly contain an $O(10-d, 10-d)$ vector $\xi_A$ (or $\xi_{+A}$) which potentially plays the role of $Z_A$. There, the Leibniz identities Eqs. (2.8)–(2.10) appear as some components of the quadratic constraints studied in [28, 32, 33].

# 3 Jacobi–Lie $T$-duality

In [20–22], the Poisson–Lie $T$-duality/$T$-plurality has been proven to be a symmetry of DFT. As a natural extension, non-Abelian $U$-duality associated with EDA has been discussed in [4–10, 16], and several examples of the non-Abelian $U$-duality have been found in [11]. Here, we show that the non-Abelian duality based on a $DD^+$, i.e., the Jacobi–Lie $T$-plurality, is a symmetry of the DFT equations of motion.

## 3.1 Generalized fluxes

In type II DFT, the bosonic fields in the NS–NS sector are the generalized metric and the DFT dilaton

$$\mathcal{H}_{MN} \equiv \begin{pmatrix} g_{mn} - B_{mp} g^{pq} B_{qn} & B_{mp} g^{pn} \\ -g^{mp} B_{pn} & g^{mn} \end{pmatrix}, \qquad e^{-2d} \equiv \sqrt{|\det g_{mn}|}\, e^{-2\Phi}, \tag{3.1}$$

and the R–R fields can be described as an $O(D,D)$ spinor $|F\rangle$. By making a certain ansatz for these bosonic fields, we show the covariance of the equations of motion under the Jacobi–Lie $T$-plurality, which is an $O(D,D)$ rotation discussed below.

Let us begin with a simple case where the R–R fields and the spectator fields $y^\mu$ (which do not transform under the $O(D,D)$ rotation) are not present. We consider an ansatz for the NS–NS sector fields,

$$\mathcal{H}_{MN}(x) = \mathcal{E}_M{}^A(x)\mathcal{E}_N{}^B(x)\hat{\mathcal{H}}_{AB}, \qquad e^{-2d(x)} = e^{-2\varphi(x)} e^{-\Delta(x)} \tilde{\sigma}^{D-\frac{3}{2}} |\det \ell_m^a(x)|, \tag{3.2}$$

where $\hat{\mathcal{H}}_{AB}$ is constant, $\varphi(x)$ is a certain function, and we have defined

$$\mathcal{E}_A{}^M \equiv e^{\omega(x)} E_A{}^M(x) = e^{\omega} \begin{pmatrix} e_a^m & 0 \\ -\pi^{ac} e_c^m & e^{-2\omega} r_m^a \end{pmatrix} \in O(D,D). \tag{3.3}$$

When the target space is of this form, this background is called Jacobi–Lie symmetric.

If we parameterize the constant matrix $\hat{\mathcal{H}}_{AB}$ as

$$\hat{\mathcal{H}}_{AB} \equiv \begin{pmatrix} \hat{g}_{ab} & -(\hat{g}\,\hat{\beta})_a{}^b \\ (\hat{\beta}\,\hat{g})^a{}_b & (\hat{g}^{-1} - \hat{\beta}\,\hat{g}\,\hat{\beta})^{ab} \end{pmatrix}, \tag{3.4}$$

by comparing the parameterization (3.1) with (3.2), the metric and the $B$-field can be expressed as $g_{mn} + B_{mn} = \mathcal{E}_{mn}$ where $\mathcal{E}_{mn}$ is the inverse matrix of

$$\mathcal{E}^{mn} \equiv e^{2\omega}\big(\hat{g} + \hat{\beta} + \pi\big)^{ab} e_a^m e_b^n. \tag{3.5}$$

They can be also expressed as

$$g_{mn} + B_{mn} = e^{-2\omega} R_{ab} r_m^a r_n^b, \qquad (R_{ab}) \equiv (\hat{g}^{ab} + \hat{\beta}^{ab} + \pi^{ab})^{-1}. \tag{3.6}$$

The standard dilaton $\Phi$ can be found as

$$e^{-2\Phi} = \sqrt{|\det \hat{g}_{ab}|}\, e^{-2\varphi(x)}\, e^{(D-1)\Delta}\, \tilde{\sigma}^{\frac{D-3}{2}}\, |\det(\hat{g}^{ab} + \hat{\beta}^{ab} + \pi^{ab})\det(a_a{}^b)|. \tag{3.7}$$

The structure constants $Z_A$, which are not present in the Poisson–Lie $T$-duality, produce the overall factor $e^{-2\omega}$ both in the metric and the $B$-field. We find that $\mathcal{E}_{mn}$ satisfies

$$\pounds_{v_a}\mathcal{E}_{mn} + 2\,Z_a\,\mathcal{E}_{mn} = -\tilde{\sigma}^{-1}\big(f_a{}^{bc} + 2\,\delta_a^b\,Z^c - 2\,\delta_a^c\,Z^b\big)\mathcal{E}_{mp}\, v_b^p\, v_c^q\, \mathcal{E}_{qn}. \tag{3.8}$$

Here, let us comment on the difference between our proposal and the one studied in [17]. In [17], the metric and the $B$-field are identified as

$$g_{mn} + B_{mn} = E_{mn}, \qquad E_{mn} \equiv R_{ab}\, r_m^a\, r_n^b\, \big(= e^{2\omega}\, \mathcal{E}_{mn}\big), \tag{3.9}$$

for which we have

$$\pounds_{v_a} E_{mn} = -e^{-2\Delta}\big(f_a{}^{bc} + 2\,\delta_a^b\,Z^c - 2\,\delta_a^c\,Z^b\big)E_{mp}\, v_b^p\, v_c^q\, E_{qn}. \tag{3.10}$$

The difference is only in the overall factor $e^{2\omega}$. Below, we show the covariance of the equations of motion under the Jacobi–Lie $T$-plurality by adopting the former choice $g_{mn} + B_{mn} = \mathcal{E}_{mn}$ and using the dilaton (3.7).

The generalized fluxes associated with $\mathcal{E}_A{}^M$ are defined as

$$\begin{aligned} \mathcal{F}_{ABC} &\equiv 3\,\mathcal{W}_{[ABC]}, \qquad \mathcal{F}_A \equiv \mathcal{W}^B{}_{AB} + 2\,\mathcal{D}_A d, \\ \mathcal{W}_{ABC} &\equiv -\mathcal{D}_A \mathcal{E}_B{}^M\, \mathcal{E}_{MC}, \qquad \mathcal{D}_A \equiv \mathcal{E}_A{}^M\, \partial_M. \end{aligned} \tag{3.11}$$

Using the algebraic and the differential identities, we find

$$\begin{aligned} &\mathcal{F}_{ABC} = e^{\omega} F_{ABC}, \quad \mathcal{F}_A = \mathcal{E}_A{}^M F_M, \\ &F_M = 2\,\partial_M d + \begin{pmatrix} \partial_m \ln|\det \ell_m^a| - \partial_m \Delta \\ -\tilde{\sigma}^{-1} f_b{}^{ba} v_a^m + \tilde{\partial}^m \ln \tilde{\sigma}^{D-\frac{3}{2}} \end{pmatrix} = 2\,\partial_M \varphi + \begin{pmatrix} 0 \\ -\frac{f_b{}^{ba} v_a^m}{\tilde{\sigma}} \end{pmatrix}, \end{aligned} \tag{3.12}$$

where $F_{ABC}$ is the one given in (2.6) and we have used Eq. (3.2). When $f_b{}^{ba}$ does not vanish, we can remove the last term by making a replacement [21]

$$\partial_M d \to \partial_M d + \mathbf{X}_M, \qquad \mathbf{X}_M = \big(0, \tfrac{1}{2\tilde{\sigma}} f_b{}^{ba} v_a^m\big), \tag{3.13}$$

and then the single-index flux becomes

$$\mathcal{F}_A = e^{\omega} F_A, \qquad F_A \equiv E_A{}^M F_M, \qquad F_M \equiv 2\,\partial_M \varphi. \tag{3.14}$$

In the following, we suppose that $F_A$ is constant.

## 3.2 Covariance of the equations of motion

In general, the equations of motion of DFT are given by

$$\mathcal{R} = 0, \qquad \mathcal{G}^{AB} = 0. \tag{3.15}$$

Here, $\mathcal{R}$ and $\mathcal{G}^{AB}$, under the section condition, can be expressed as

$$\mathcal{R} \equiv \hat{\mathcal{H}}^{AB} \left(2 \mathcal{D}_A \mathcal{F}_B - \mathcal{F}_A \mathcal{F}_B\right) + \tfrac{1}{12} \hat{\mathcal{H}}^{AD} \left(3 \eta^{BE} \eta^{CF} - \hat{\mathcal{H}}^{BE} \hat{\mathcal{H}}^{CF}\right) \mathcal{F}_{ABC} \mathcal{F}_{DEF}, \tag{3.16}$$

$$\mathcal{G}^{AB} \equiv 2 \hat{\mathcal{H}}^{D[A} \mathcal{D}^{B]} \mathcal{F}_D - \tfrac{1}{2} \hat{\mathcal{H}}^{DE} (\eta^{AF} \eta^{BG} - \hat{\mathcal{H}}^{AF} \hat{\mathcal{H}}^{BG})(\mathcal{F}_D - \mathcal{D}_D) \mathcal{F}_{EFG}$$
$$- \hat{\mathcal{H}}_E{}^{[A} (\mathcal{F}_D - \mathcal{D}_D) \mathcal{F}^{B]DE} + \tfrac{1}{2} (\eta^{CE} \eta^{DF} - \hat{\mathcal{H}}^{CE} \hat{\mathcal{H}}^{DF}) \hat{\mathcal{H}}^{G[A} \mathcal{F}_{CD}{}^{B]} \mathcal{F}_{EFG}. \tag{3.17}$$

In our setup, we find important relations

$$\mathcal{D}_D \mathcal{F}_{ABC} = \mathrm{e}^{2\omega} Z_D F_{ABC}, \qquad \mathcal{D}_D \mathcal{F}_A = \mathrm{e}^{2\omega} Z_D F_A, \tag{3.18}$$

and we obtain

$$\mathcal{R} = \mathrm{e}^{2\omega} R, \qquad \mathcal{G}^{AB} = \mathrm{e}^{2\omega} G^{AB}, \tag{3.19}$$

where $R$ and $G^{AB}$ are constants of the form

$$R \equiv \hat{\mathcal{H}}^{AB} \left(2 Z_A F_B - F_A F_B\right) + \tfrac{1}{12} \hat{\mathcal{H}}^{AD} \left(3 \eta^{BE} \eta^{CF} - \hat{\mathcal{H}}^{BE} \hat{\mathcal{H}}^{CF}\right) F_{ABC} F_{DEF}, \tag{3.20}$$

$$G^{AB} \equiv 2 \hat{\mathcal{H}}^{D[A} F^{B]} F_D - \tfrac{1}{2} \hat{\mathcal{H}}^{DE} (\eta^{AF} \eta^{BG} - \hat{\mathcal{H}}^{AF} \hat{\mathcal{H}}^{BG})(F_D - Z_D) F_{EFG}$$
$$- \hat{\mathcal{H}}_E{}^{[A} (F_D - Z_D) F^{B]DE} + \tfrac{1}{2} (\eta^{CE} \eta^{DF} - \hat{\mathcal{H}}^{CE} \hat{\mathcal{H}}^{DF}) \hat{\mathcal{H}}^{G[A} F_{CD}{}^{B]} F_{EFG}. \tag{3.21}$$

Then the equations of motion simply become $R = 0$ and $G^{AB} = 0$, which are manifestly covariant under the $\mathrm{O}(D,D)$ rotation

$$\begin{aligned} F_{ABC} &\to C_A{}^D C_B{}^E C_C{}^F F_{DEF}, & Z_A &\to C_A{}^B Z_B, \\ \hat{\mathcal{H}}_{AB} &\to C_A{}^C C_B{}^D \hat{\mathcal{H}}_{CD}, & F_A &\to C_A{}^B F_B. \end{aligned} \tag{3.22}$$

The transformations in the first line are equivalent to a redefinition of generators

$$T_A \to C_A{}^B T_B, \tag{3.23}$$

while those in the second line determine the transformation rules of $\hat{\mathcal{H}}_{AB}$ and $\varphi$. This $\mathrm{O}(D,D)$ symmetry is the Jacobi–Lie $T$-plurality and is a manifest symmetry of DFT.

For later convenience, let us also find the transformation rule of the generalized Ricci tensor $\mathcal{S}_{MN}$. We define the (constant) double vielbein $V_A{}^{\mathcal{B}} \equiv (V_a{}^{\mathcal{B}}, V_{\bar{a}}{}^{\mathcal{B}}) \in \mathrm{O}(D,D)$ and its inverse $V_A{}^{\mathcal{B}}$ through

$$\hat{\mathcal{H}}_{AB} = V_A{}^{\mathcal{A}} V_B{}^{\mathcal{B}} \hat{\mathcal{H}}_{\mathcal{AB}}, \qquad \eta_{AB} = V_A{}^{\mathcal{A}} V_B{}^{\mathcal{B}} \eta_{\mathcal{AB}}, \qquad V_A{}^{\mathcal{C}} V_{\mathcal{C}}{}^B = \delta_A^B, \tag{3.24}$$

where

$$(\hat{\mathcal{H}}_{\mathcal{AB}}) \equiv \begin{pmatrix} \eta_{ab} & 0 \\ 0 & \eta_{\bar{a}\bar{b}} \end{pmatrix}, \qquad (\eta_{\mathcal{AB}}) \equiv \begin{pmatrix} \eta_{ab} & 0 \\ 0 & -\eta_{\bar{a}\bar{b}} \end{pmatrix}, \tag{3.25}$$

and $\eta_{ab} \equiv \eta_{\bar{a}\bar{b}} \equiv \mathrm{diag}(-1, 1, \ldots, 1)$. We suppose that the double vielbein is transformed as

$$V_A{}^{\mathcal{B}} \to C_A{}^C V_C{}^{\mathcal{B}}, \tag{3.26}$$

under the Jacobi–Lie $T$-duality, and then the transformation rule for

$$\mathcal{G}^{\mathcal{AB}} \equiv V_A{}^{\mathcal{A}} V_B{}^{\mathcal{B}} \mathcal{G}^{AB} , \tag{3.27}$$

is found as

$$e^{-2\omega'} \mathcal{G}'^{\mathcal{AB}} = e^{-2\omega} \mathcal{G}^{\mathcal{AB}} . \tag{3.28}$$

We find that the only non-vanishing components of $\mathcal{G}^{\mathcal{AB}}$ are $\mathcal{G}^{a\bar{b}}$, and using these, we can express the generalized Ricci tensor as

$$\mathcal{S}_{MN} = \left(\mathcal{E}_{MA}\mathcal{E}_{NB} + \mathcal{E}_{NA}\mathcal{E}_{MB}\right) V_c{}^A V_{\bar{d}}{}^B \mathcal{G}^{c\bar{d}} . \tag{3.29}$$

Then, using (3.28), we find the transformation rule of the generalized Ricci tensor $\mathcal{S}_{MN}$ as

$$e^{-2\omega'} \mathcal{E}'^M_A \mathcal{E}'^N_B \mathcal{S}'_{MN} = e^{-2\omega} C_A{}^C C_B{}^D \mathcal{E}_C{}^M \mathcal{E}_D{}^N \mathcal{S}_{MN} . \tag{3.30}$$

Namely, under the Jacobi–Lie $T$-plurality, or a local O$(D,D)$ rotation of the generalized metric,

$$\mathcal{H}_{MN}(x) \to \mathcal{H}'_{MN}(x') = \left[h\,\mathcal{H}(x)\,h^{\mathrm{t}}\right]_{MN} , \qquad h_M{}^N \equiv \mathcal{E}'_M{}^A(x')\,C_A{}^B\,\mathcal{E}_B{}^N(x) , \tag{3.31}$$

the generalized Ricci tensor transforms as

$$\mathcal{S}_{MN}(x) \to \mathcal{S}'_{MN}(x') = e^{2(\omega'-\omega)}\left[h\,\mathcal{S}(x)\,h^{\mathrm{t}}\right]_{MN} . \tag{3.32}$$

Unlike the case of the Poisson–Lie $T$-duality, the generalized Ricci tensor is transformed by a local O$(D,D) \times \mathbb{R}^+$ rotation. As we discuss later, this additional $\mathbb{R}^+$ transformation makes the transformation rule of the R–R fields slightly non-trivial.

**Comments on a subtle issue**

Here, we comment on an issue that may arise in the presence of $Z^a$ and $f_b{}^{ba}$.

Firstly, we consider the case where two vectors $I \equiv \frac{1}{2} f_b{}^{ba} v_a$ and $Z = Z^a v_a$ are proportional to each other (which includes the case where $I = 0$ or $Z = 0$). Since $Z$ is a Killing vector field, we can choose a coordinate system such that $Z = c_Z \partial_w$ and $I = c_I \partial_w$ (where $c_Z$ and $c_I$ are constants). In such a coordinate system, recalling $\tilde{\partial}^m \tilde{\sigma} = -2 Z^m$, we find

$$\tilde{\sigma} = c_0 - 2 c_Z \tilde{w} , \tag{3.33}$$

where $c_0$ is a constant. Now we consider the following three cases.

1. $c_I = 0$ and $c_Z = 0$

   In this case, the shift (3.13) is not necessary, and we can choose $\tilde{\sigma} = 1$ by a redefinition of $\varphi(x)$. Then the metric and the $B$-field are independent of the dual coordinates. The section condition is satisfied if $\varphi$ satisfies $\partial_M\partial^M\varphi = \partial_M\varphi\,\partial^M\varphi = \partial_M\varphi\,\partial^M d = \partial_M\varphi\,\partial^M\mathcal{H}_{PQ} = 0$. By recalling Eq. (3.14), they are equivalent to

$$\mathcal{F}_A \mathcal{F}^A = \mathcal{F}^A \mathcal{D}_A d = \mathcal{F}^A \mathcal{D}_A \mathcal{H}_{MN} = 0 . \tag{3.34}$$

   In particular, if $\varphi$ is independent of the dual coordinates, the DFT solution corresponds to a solution of the usual supergravity.

2. $c_I \neq 0$ and $c_Z = 0$

   Again we can choose $\tilde{\sigma} = 1$, and then the metric and the $B$-field are independent of the dual coordinates. The shift (3.13) corresponds to introducing the dual-coordinate dependence into the dilaton [34, 35] $d \to d + c_I \tilde{w}$. Namely, the dilaton becomes

$$e^{-2 d(x)} = e^{-2\varphi(x)} e^{-\Delta(x) - 2 c_I \tilde{w}} |\det \ell_m^a|. \tag{3.35}$$

   The section condition is satisfied if $\pounds_I\big(e^{-2\varphi(x)}|\det \ell_m^a|\big) = 0$ and Eq. (3.34) are satisfied. In this case, for example if $\varphi$ is independent of the dual coordinates, the DFT solution corresponds to a solution of the generalized supergravity equations of motion [35–37].

3. $c_Z \neq 0$

   We find $\frac{1}{2\tilde{\sigma}} f_b{}^{ba} v_a = \frac{c_I}{c_0 - 2 c_Z \tilde{w}} \partial_w$ and then the shift (3.13) corresponds to the shift

$$d \to d - \frac{c_I}{2 c_Z} \ln(c_0 - 2 c_Z \tilde{w}). \tag{3.36}$$

   Then, the dilaton becomes

$$e^{-2 d(x)} = e^{-2\varphi(x)} e^{-\Delta(x)} (c_0 - 2 c_Z \tilde{w})^{D - \frac{3}{2} + \frac{c_I}{c_Z}} |\det \ell_m^a|. \tag{3.37}$$

   Here, the Leibniz identities ensures $\pounds_Z |\det \ell_m^a| = 0$, and the section condition is satisfied if $\pounds_Z \varphi = 0$ ($\Leftrightarrow Z^a F_a = 0 = \pi^{ab} Z_a F_b$) and Eq. (3.34) are satisfied. Even if this is a solution of DFT, this does not correspond to a solution of the usual (or the generalized) supergravity because the metric and the $B$-field depend on the dual coordinates.

If $F_A = 0$ (or $\varphi = 0$), we do not need to care about the section condition: only the second case requires a non-trivial relation $f_b{}^{ba} f_{ac}{}^c = 0$ ($\Leftrightarrow \pounds_I |\det \ell_m^a| = 0$). When $F_A \neq 0$, a non-trivial issue can arise. Even if $F_A = 2 E_A{}^M \partial_M \varphi$ is satisfied in the original frame, after the Jacobi–Lie $T$-plurality ($F_A \to F_A' = C_A{}^B F_B$), it may be possible that there is no function $\varphi'$ that satisfies $F_A' = 2 E_A'^M \partial_M' \varphi'$.[4] In such a case, we cannot find the DFT dilaton, and it does not correspond to a solution of the usual DFT.

Secondly, let us consider the problematic case where two vector fields $I$ and $Z$ are linearly dependent. Using the Leibniz identities (2.8)–(2.10), we can show that they commute with each other $[I, Z] = 0$. Since $Z$ is a Killing vector field, if we suppose that $I$ is also a Killing vector field, we can choose a coordinate system such that $Z = \partial_w$ and $I = \partial_z$. In such a coordinate system, we find $\tilde{\sigma} = c_0 - 2\tilde{w}$ ($c_0$: arbitrary constant) and we also find $\frac{1}{2\tilde{\sigma}} f_b{}^{ba} v_a = \frac{1}{c_0 - 2\tilde{w}} \partial_z$. After the sift (3.13), the derivative of the dilaton becomes

$$\partial_M d = \mathcal{Y}_M, \qquad \mathcal{Y}_M \equiv \tfrac{1}{2} \partial_M \big[\Delta - \ln(\tilde{\sigma}^{D - \frac{3}{2}} |\det \ell_m^a|)\big] + \tfrac{1}{2} F_M + \begin{pmatrix} 0 \\ \frac{1}{c_0 - 2\tilde{w}} \delta_z^m \end{pmatrix}, \tag{3.38}$$

where $F_M$ is defined in (3.14). If there exists a function $\zeta$ that satisfies $\mathcal{Y}_M = \partial_M \zeta$, we can obtain the DFT dilaton as $d = \zeta$. However, in general, the vector field $\mathcal{Y}_M$ satisfies $\partial_{[M} \mathcal{Y}_{N]} \neq 0$ and then there is no solution for $\mathcal{Y}_M = \partial_M \zeta$. In particular, when $F_A = 0$, due to the existence of the last term of $\mathcal{Y}_M$, we find $\tilde{\partial}^{[w} \mathcal{Y}^{z]} = -(c_0 - 2\tilde{w})^{-2} \neq 0$. Therefore, only when a flux $F_A$ is introduced such that the last term of $\mathcal{Y}_M$ is canceled out, we can obtain the DFT dilaton. Moreover, the section condition, such as $\partial^M \partial_M d = \partial_M d\, \partial^M \mathcal{H}_{PQ} = 0$, are also not ensured. When $I$ is not a Killing vector field, the situation will be worse. From the above consideration, we conclude that it is difficult to construct a DFT solution when $f_b{}^{ba}$ and $Z^a$ are linearly independent. In section 3.4.1, we show a concrete example of this type, where the DFT dilaton $d$ cannot be determined and the section condition is broken by $\partial_M d$.

---

[4]In the Poisson–Lie $T$-plurality, there is a prescription to find $\varphi'(x')$, which is based on a coordinate transformation on the Drinfel'd double [3]. However, unlike a Lie algebra which can be exponentiated to a Lie group, it is not clear how to globally extend the Leibniz algebra DD$^+$ to a group-like space. Then the procedure of [3] does not work and the function $\varphi'$ needs to be found by solving the differential equation $F_A' = 2 E_A'^M \partial_M' \varphi'$.

### 3.3 An example without Ramond–Ramond flux

Let us consider an eight-dimensional Leibniz algebra with

$$f_{12}{}^2 = -1, \quad f_{12}{}^3 = 1, \quad f_{13}{}^3 = -1, \quad Z_1 = -2, \tag{3.39}$$

which is a direct sum of the six-dimensional Leibniz algebra $((\text{IV}, -4\tilde{X}^1), (\text{I}, 0))$ of [25] and a two-dimensional Abelian algebra. Using a parameterization $g = e^{x\,T_1}\,e^{y\,T_2}\,e^{z\,T_3}\,e^{w\,T_4}$, we find

$$\begin{aligned}
v_1 &= \partial_x + y\,\partial_y + (z-y)\,\partial_z, \quad v_2 = \partial_y, \quad v_3 = \partial_z, \quad v_4 = \partial_w, \\
e_1 &= \partial_x, \quad e_2 = e^x(\partial_y - x\,\partial_z), \quad e_3 = e^x\,\partial_z, \quad e_4 = \partial_w.
\end{aligned} \tag{3.40}$$

Computing the matrix $M_A{}^B$, we find

$$\pi^{ab} = 0, \qquad \Delta = -2\,x. \tag{3.41}$$

Then, using the constant matrices

$$\hat{g}_{ab} = \begin{pmatrix} 0 & 0 & 1 & 0 \\ 0 & 1 & 0 & 0 \\ 1 & 0 & 0 & 0 \\ 0 & 0 & 0 & 1 \end{pmatrix}, \qquad \hat{\beta}^{ab} = 0, \tag{3.42}$$

we obtain a 4D metric

$$\mathrm{d}s^2 = 2\,e^{3x}\,\mathrm{d}x\,(\mathrm{d}z + x\,\mathrm{d}y) + e^{2x}\,\mathrm{d}y^2 + e^{4x}\,\mathrm{d}w^2. \tag{3.43}$$

In order to find a solution of DFT, we choose the function $\varphi$ as

$$\varphi = -\tfrac{4}{3}\,x, \tag{3.44}$$

which yields

$$F_A = \left(-\tfrac{8}{3}, 0, 0, 0, 0, 0, 0, 0\right). \tag{3.45}$$

Then the DFT dilaton and the standard dilaton become

$$e^{-2d} = e^{\frac{14x}{3}}, \qquad e^{-2\Phi} = e^{-\frac{4x}{3}}. \tag{3.46}$$

We can check that this dilaton and the metric (3.43) satisfy the equations of motion. In the following, we consider the Jacobi–Lie $T$-pluralities of this solution.

#### 3.3.1 Generalized Yang–Baxter deformation

Let us perform an $O(4,4)$ rotation $T_A \to C_A{}^B\,T_B$ with

$$C_A{}^B = \begin{pmatrix} \delta_a^b & 0 \\ r^{ab} & \delta_b^a \end{pmatrix}, \qquad r^{ab} = \begin{pmatrix} 0 & 0 & 0 & 0 \\ 0 & 0 & c & 0 \\ 0 & -c & 0 & 0 \\ 0 & 0 & 0 & 0 \end{pmatrix}. \tag{3.47}$$

The original algebra (3.39) has vanishing $f_a{}^{bc}$, but this $O(4,4)$ rotation produces the dual structure constants of the coboundary type (2.40). In the presence of $Z_A$, this type of $O(D,D)$ rotation characterized by an antisymmetric matrix $r^{ab}$ may be called the generalized Yang–Baxter deformation because the matrix $r^{ab}$ is a solution of the generalized classical Yang–Baxter equations (2.41). After this $O(4,4)$ rotation, the structure constants becomes

$$f_{12}{}^2 = -1, \quad f_{12}{}^3 = 1, \quad f_{13}{}^3 = -1, \quad f_1{}^{23} = 2\,c, \quad Z_1 = -2, \tag{3.48}$$

and this corresponds $((\text{IV}, -4\tilde{X}^1), (\text{II}, 0))$ or $((\text{IV.iii}, 4\tilde{X}^1), (\text{II}, 0))$ of [25] (accompanied by the two-dimensional Abelian algebra), for $c = 1/2$ or $c = -1/2$, respectively.[5]

Again we employ the same parametrization of the group element and the left-/right-invariant vector fields (3.40). Here, we find the Jacobi–Lie structure as

$$\pi = c\,(1 - \mathrm{e}^{-2x})\,\partial_y \wedge \partial_z\,, \tag{3.49}$$

and $\varphi$ is not changed because $\mathcal{F}_A$ is not deformed under this O(4, 4) rotation: $\mathcal{F}_A = C_A{}^B \mathcal{F}_B$. Then, we find the deformed supergravity fields as

$$\begin{aligned}
\mathrm{d}s^2 &= 2\,\mathrm{e}^{3x}\,\mathrm{d}x\,(\mathrm{d}z + x\,\mathrm{d}y) + \mathrm{e}^{2x}\,\mathrm{d}y^2 + \mathrm{e}^{4x}\,\mathrm{d}w^2 + \mathrm{e}^{2x}\,c^2\,\mathrm{d}z^2\,, \\
B_2 &= c\,\mathrm{e}^{5x}\,\mathrm{d}x \wedge \mathrm{d}y\,, \qquad \mathrm{e}^{-2\Phi} = \mathrm{e}^{-\frac{4x}{3}}\,.
\end{aligned} \tag{3.50}$$

This is again a supergravity solution for an arbitrary value of $c$.

### 3.3.2 Another Jacobi–Lie $T$-plurality

Here we consider another O(4, 4) transformation

$$C_A{}^B = \begin{pmatrix}
1 & 0 & 0 & 0 & 0 & 0 & 0 & 0 \\
0 & 0 & -1 & 0 & 0 & \frac{1}{2} & 0 & 0 \\
0 & 0 & 1 & 0 & 0 & \frac{1}{2} & 0 & 0 \\
0 & 0 & 0 & 1 & 0 & 0 & 0 & 0 \\
0 & 0 & 0 & 0 & 1 & 0 & 0 & 0 \\
0 & 1 & 0 & 0 & 0 & 0 & -\frac{1}{2} & 0 \\
0 & 1 & 0 & 0 & 0 & 0 & \frac{1}{2} & 0 \\
0 & 0 & 0 & 0 & 0 & 0 & 0 & 1
\end{pmatrix}. \tag{3.51}$$

We then obtain the algebra with

$$f_{12}{}^2 = -2\,, \quad f_{12}{}^3 = -1\,, \quad f_{13}{}^2 = -1\,, \quad f_{13}{}^3 = -2\,, \quad f_1{}^{23} = 1\,, \quad Z_1 = -2\,. \tag{3.52}$$

The six-dimensional part of this algebra is known as $((\text{VI}_2, -4\tilde{X}^1), (\text{II}, 0))$. Using the parameterization, $g = \mathrm{e}^{x\,T_1}\,\mathrm{e}^{y\,T_2}\,\mathrm{e}^{z\,T_3}\,\mathrm{e}^{w\,T_4}$, we obtain

$$\begin{aligned}
v_1 &= \partial_x + (2y + z)(\partial_y + \partial_z)\,, \quad v_2 = \partial_y\,, \quad v_3 = \partial_z\,, \quad v_4 = e_4 = \partial_w\,, \\
e_1 &= \partial_x\,, \quad e_2 = \frac{\mathrm{e}^x}{2}\big[(\mathrm{e}^{2x}+1)\,\partial_y + (\mathrm{e}^{2x}-1)\,\partial_z\big]\,, \quad e_3 = \frac{\mathrm{e}^x}{2}\big[(\mathrm{e}^{2x}-1)\,\partial_y + (\mathrm{e}^{2x}+1)\,\partial_z\big]\,.
\end{aligned} \tag{3.53}$$

We can compute several quantities as

$$\pi = x\,\partial_y \wedge \partial_z\,, \qquad \Delta = -2x\,, \qquad \varphi = -\tfrac{4}{3}x\,. \tag{3.54}$$

The associated supergravity fields are found as

$$\begin{aligned}
\mathrm{d}s^2 &= \mathrm{e}^{4x}(\mathrm{d}w^2 - x^2\,\mathrm{d}x^2) - 2\,\mathrm{e}^{3x}\,\mathrm{d}x\,(\mathrm{d}y - \mathrm{d}z) + \tfrac{1}{4}\,\mathrm{e}^{-2x}(\mathrm{d}y + \mathrm{d}z)^2\,, \\
B_2 &= \tfrac{1}{2}\,\mathrm{e}^x\,x\,\mathrm{d}x \wedge (\mathrm{d}y + \mathrm{d}z)\,, \qquad \mathrm{e}^{-2\Phi} = \mathrm{e}^{\frac{2x}{3}}\,,
\end{aligned} \tag{3.55}$$

and this is a solution of the supergravity.

---

[5] The algebra with $c > 0$ or $c < 0$ can be mapped to to the one with $c = 1/2$ or $c = -1/2$, respectively.

### 3.3.3 Jacobi–Lie $T$-duality

To provide an example with $Z^a \neq 0$, let us consider the $T$-dual of the previous example. The non-vanishing structure constants are

$$f_{23}{}^1 = 1, \quad f_2{}^{12} = -2, \quad f_3{}^{12} = -1, \quad f_2{}^{13} = -1, \quad f_3{}^{13} = -2, \quad Z^1 = -2, \tag{3.56}$$

and the constant metric $\hat{\mathcal{H}}_{AB}$ and the flux $F_A$ become

$$
\hat{\mathcal{H}}_{AB} = \begin{pmatrix}
0 & -\frac{1}{2} & \frac{1}{2} & 0 & 0 & 0 & 0 & 0 \\
-\frac{1}{2} & 1 & 1 & 0 & 0 & 0 & 0 & 0 \\
\frac{1}{2} & 1 & 1 & 0 & 0 & 0 & 0 & 0 \\
0 & 0 & 0 & 1 & 0 & 0 & 0 & 0 \\
0 & 0 & 0 & 0 & 0 & -1 & 1 & 0 \\
0 & 0 & 0 & 0 & -1 & \frac{1}{4} & \frac{1}{4} & 0 \\
0 & 0 & 0 & 0 & 1 & \frac{1}{4} & \frac{1}{4} & 0 \\
0 & 0 & 0 & 0 & 0 & 0 & 0 & 1
\end{pmatrix}, \qquad
F_A = \begin{pmatrix}
0 \\ 0 \\ 0 \\ 0 \\ -\frac{8}{3} \\ 0 \\ 0 \\ 0
\end{pmatrix}. \tag{3.57}
$$

Since we find $\frac{1}{2} f_b{}^{ba} = -Z^a$, this example corresponds to the third case discussed around Eq. (3.37). By using a parameterization $g = e^{x T_1} e^{y T_2} e^{z T_3} e^{w T_4}$, we find

$$
\begin{aligned}
e_1 &= \partial_x, \quad e_2 = \partial_y, \quad e_3 = \partial_z - y\,\partial_x, \quad e_4 = \partial_w, \\
v_1 &= \partial_x, \quad v_2 = \partial_y - z\,\partial_x, \quad v_3 = \partial_z, \quad v_4 = \partial_w,
\end{aligned} \tag{3.58}
$$

$$
\pi = \partial_x \wedge \left[ (2y - z)\,\partial_y - (y - 2z)\,\partial_z + 4w\,\partial_w \right], \qquad \tilde{\sigma} = c_0 + 4\tilde{x}. \tag{3.59}
$$

Then, we can easily compute the generalized frame fields $E_A{}^M$ and $\mathcal{E}_A{}^M$. The resulting generalized metric shows that the metric and the $B$-fields are

$$
g_{mn} = \tilde{\sigma} \begin{pmatrix}
0 & \frac{2}{9(y-z)^2-4} & -\frac{2}{9(y-z)^2-4} & 0 \\
& \frac{4(4w^2+y^2-4yz+4z^2-1)}{9(y-z)^2-4} & \frac{2(-8w^2+4y^2-10yz+y+4z^2-2)}{9(y-z)^2-4} & \frac{12w(z-y)}{9(y-z)^2-4} \\
& & \frac{4(4w^2-4yz+y(4y-1)+z^2-1)}{9(y-z)^2-4} & \frac{12w(y-z)}{9(y-z)^2-4} \\
& & & 1
\end{pmatrix},
$$

$$
B_{mn} = \tilde{\sigma} \begin{pmatrix}
0 & \frac{3z-3y}{9(y-z)^2-4} & \frac{3(y-z)}{9(y-z)^2-4} & 0 \\
& 0 & \frac{y(3y-3z-4)-4z}{9(y-z)^2-4} & -\frac{8w}{9(y-z)^2-4} \\
& & 0 & \frac{8w}{9(y-z)^2-4} \\
& & & 0
\end{pmatrix}. \tag{3.60}
$$

The flux $F_A$ shows that $\varphi(x) = -\frac{1}{3}\ln\tilde{\sigma}$, and by using the formula (3.37), the DFT dilaton is found as

$$d = -\tfrac{13}{12}\ln\tilde{\sigma}. \tag{3.61}$$

This dilaton together with the generalized metric (3.60) satisfies the DFT equations of motion. Since the metric and the $B$-field have the dual-coordinate dependence through the overall factor $\tilde{\sigma}$, this is not a solution of the usual (or the generalized) supergravity. However, the section condition is not broken and can be mapped to a DFT solution that does not depend on dual coordinates.

### 3.4 Another example without Ramond–Ramond flux

To provide a problematic example, let us consider

$$f_{12}{}^2 = -1, \quad f_{12}{}^3 = -1, \quad f_{13}{}^2 = -1, \quad f_{13}{}^3 = -1, \quad Z_2 = -\frac{1}{2}, \quad Z_3 = \frac{1}{2}, \tag{3.62}$$

which corresponds to the $T$-dual of $((I, 0), (III, -(\tilde{X}^2 - \tilde{X}^3)))$. Using a parameterization, $g = e^{x T_1} e^{(y+z) T_2} e^{(y-z) T_3}$ we find

$$
e_a{}^m = \begin{pmatrix} 1 & 0 & 0 \\ 0 & \frac{e^{2x}}{2} & \frac{1}{2} \\ 0 & \frac{e^{2x}}{2} & -\frac{1}{2} \end{pmatrix}, \qquad v_a{}^m = \begin{pmatrix} 1 & 2y & 0 \\ 0 & \frac{1}{2} & \frac{1}{2} \\ 0 & \frac{1}{2} & -\frac{1}{2} \end{pmatrix},
$$
$$
e^{-2\Delta} = e^{2z}, \qquad \pi^{mn} = 0,
$$
(3.63)

and then by introducing

$$
\hat{g}_{ab} = \begin{pmatrix} -2 & 0 & 0 \\ 0 & \frac{1}{4} & 0 \\ 0 & 0 & \frac{1}{4} \end{pmatrix}, \qquad \hat{\beta}^{ab} = 0,
$$
(3.64)

we obtain a 3D metric

$$
\mathrm{d}s^2 = \frac{1}{2} e^{2z} \big(e^{-4x} \, \mathrm{d}y^2 + \mathrm{d}z^2 - 4 \, \mathrm{d}x^2\big).
$$
(3.65)

This is a flat Minkowski space and is a trivial solution of supergravity. In order to realize $\Phi = 0$, we introduce $\varphi(x) = x - z$. We then find

$$
F_A = (2, -1, 1, 0, 0, 0).
$$
(3.66)

### 3.4.1 A problematic example

Now we perform the Jacobi–Lie $T$-duality, $T^a \leftrightarrow T_a$. The resulting DD$^+$ has the structure constants

$$
f_2{}^{12} = -1, \quad f_3{}^{12} = -1, \quad f_2{}^{13} = -1, \quad f_3{}^{13} = -1, \quad Z^2 = -\frac{1}{2}, \quad Z^3 = \frac{1}{2},
$$
(3.67)

and the flux $F_A$ becomes

$$
F_A = (0, 0, 0, 2, -1, 1).
$$
(3.68)

In this case, $\frac{1}{2} f_b{}^{ba}$ and $Z^a$ are linearly independent, which is problematic as we have discussed. Using a parameterization $g = e^{x T_1} e^{y T_2} e^{z T_3}$, we can straightforwardly compute the generalized metric as

$$
\mathcal{H}_{MN} = \begin{pmatrix} -\frac{\tilde{\sigma}}{2} & 0 & 0 & 0 & -\frac{x+y+z}{2} & \frac{x-y-z}{2} \\ & 4\tilde{\sigma} & 0 & -4(x+y+z) & 0 & -4(y+z) \\ & & 4\tilde{\sigma} & 4(x-y-z) & 4(y+z) & 0 \\ & & & \frac{8x^2+8(y+z)^2-2}{\tilde{\sigma}} & \frac{4(y+z)(x-y-z)}{\tilde{\sigma}} & \frac{4(y+z)(x+y+z)}{\tilde{\sigma}} \\ & & & & \frac{-2x^2-4x(y+z)+14(y+z)^2+1}{4(c_0+\tilde{y}-\tilde{z})} & \frac{(x-y-z)(x+y+z)}{2\tilde{\sigma}} \\ & & & & & \frac{-2x^2+4x(y+z)+14(y+z)^2+1}{4\tilde{\sigma}} \end{pmatrix},
$$
(3.69)

where $\tilde{\sigma} \equiv c_0 + \tilde{y} - \tilde{z}$. One can check that this satisfies the section condition, $\partial^P \partial_P \mathcal{H}_{MN} = 0$ and $\partial^R \mathcal{H}_{MN} \partial_R \mathcal{H}_{PQ} = 0$, and there is no problem at this stage.

The problem is related to the dilaton. By using $\Delta(x) = 0$, $\tilde{\sigma} = c_0 + \tilde{y} - \tilde{z}$, and $|\det \ell_m^a| = 1$, the general formula (3.2) gives

$$
e^{-2 d(x)} = e^{-2\varphi(x)} (c_0 + \tilde{y} - \tilde{z})^{\frac{3}{2}}.
$$
(3.70)

By considering the shift (3.13) and using $v_a^m = \delta_a^m$, the derivative of the DFT dilaton becomes

$$
\partial_M d = \mathcal{Y}_M, \qquad \mathcal{Y}_M = \partial_M \ln(c_0 + \tilde{y} - \tilde{z})^{-\frac{3}{4}} + \frac{1}{2} F_M + \Big(0, \tfrac{1}{c_0+\tilde{y}-\tilde{z}} \delta_1^m\Big).
$$
(3.71)

We can easily compute $\frac{1}{2}F_M = \frac{1}{2}E_M{}^A F_A = \frac{1}{2(c_0+\tilde{y}-\tilde{z})}(0,0,0,2,-1,1)$, and then, substituting the generalized metric $\mathcal{H}_{MN}$ and

$$\partial_M d = \mathcal{Y}_M = \frac{1}{c_0+\tilde{y}-\tilde{z}}\left(0,0,0,2,-\tfrac{5}{4},\tfrac{5}{4}\right), \tag{3.72}$$

into the equations of motion, we find that the DFT equations of motion are indeed satisfied. A problem is that the section condition is broken by the DFT dilaton,

$$\partial_P d\, \partial^P \mathcal{H}_{MN} \neq 0. \tag{3.73}$$

Another problem is that we cannot find the DFT dilaton $d$ that solves the differential equation (3.72). Consequently, this configuration cannot be regarded as a solution of DFT.

## 3.5 Ramond–Ramond fields

We here introduce the R–R fields by considering the case $D = 10$. In the presence of the R–R fields, the equations of motion for the generalized metric and the DFT dilaton become

$$\mathcal{R} = 0, \qquad \mathcal{S}_{MN} = \mathcal{E}_{MN}, \tag{3.74}$$

where $\mathcal{E}_{MN}$ denotes the energy-momentum tensor of the R–R fields. Obviously, if we transform the energy-momentum tensor as

$$\mathrm{e}^{-2\omega}\,\mathcal{E}_A{}^M\,\mathcal{E}_B{}^N\,\mathcal{E}_{MN} = \mathrm{e}^{-2\omega'}\,\mathcal{E}'_A{}^M\,\mathcal{E}'_B{}^N\,\mathcal{E}'_{MN}, \tag{3.75}$$

the equations of motion for the generalized metric transform covariantly as

$$\mathrm{e}^{-2\omega}\,\mathcal{E}_A{}^M\,\mathcal{E}_B{}^N\left(\mathcal{S}_{MN}-\mathcal{E}_{MN}\right) = \mathrm{e}^{-2\omega'}\,\mathcal{E}'_A{}^M\,\mathcal{E}'_B{}^N\left(\mathcal{S}'_{MN}-\mathcal{E}'_{MN}\right). \tag{3.76}$$

By using the results of the Poisson–Lie $T$-duality [20–22], we can easily see that the transformation rule (3.75) can be realized by using the ansatz

$$|F\rangle = \sqrt{\det(\mathrm{e}^\omega\, e_a{}^m)}\,\mathrm{e}^{-d(x)}\,\mathrm{e}^\omega\, S_U|\hat{\mathcal{F}}\rangle, \tag{3.77}$$

where $U \equiv (\mathcal{E}_M{}^A)$ and $S_U$ is a matrix representation of $U$ in the spinor representation (see [22] for our convention). The presence of $\mathrm{e}^\omega$ is the only difference from the Poisson–Lie $T$-duality. The $O(10,10)$ spinor $|\hat{\mathcal{F}}\rangle$ is constant, and in type IIA/IIB theory, it can be expanded as

$$|\hat{\mathcal{F}}\rangle = \sum_{p\,:\,\text{even/odd}} \frac{1}{p!}\,\hat{\mathcal{F}}_{a_1\cdots a_p}\Gamma^{a_1\cdots a_p}|0\rangle, \tag{3.78}$$

where $|0\rangle$ is the Clifford vacuum satisfying $\Gamma_a|0\rangle = 0$. Under the ansatz (3.77), the equations of motion of the R–R fields become the algebraic relation

$$\left(\tfrac{1}{3!}\Gamma^{ABC}F_{ABC} - \tfrac{1}{2}\Gamma^A F_A + \Gamma^A Z_A\right)|\hat{\mathcal{F}}\rangle = 0. \tag{3.79}$$

When we consider an $O(D,D)$ rotation (3.22), by rotating the constant spinor $|\hat{\mathcal{F}}\rangle$ also as

$$|\hat{\mathcal{F}}\rangle \to S_C|\hat{\mathcal{F}}\rangle \qquad \left(\Gamma^B C_B{}^A = S_C\,\Gamma^A S_C^{-1}\right), \tag{3.80}$$

the equations of motion are manifestly covariant. This shows that the whole DFT equations of motion are covariant under the Jacobi–Lie $T$-plurality. When the supergravity fields have the form (3.2) and (3.77), we call the background the Jacobi–Lie symmetric.

For convenience, let us also express (3.77) in terms of the differential form. By using a polyform

$$
F \equiv \sum_{p : \text{even/odd}} \frac{1}{p!} F_{m_1 \cdots m_p} \, dx^{m_1} \wedge \cdots \wedge dx^{m_p} \,,
\tag{3.81}
$$

in type IIA/IIB theory, we have

$$
F = e^{-\varphi(x)} e^{-(p - \frac{D+1}{2}) \Delta} \, \tilde{\sigma}^{\frac{D+2p-5}{4}} |\det a_a{}^b|^{\frac{1}{2}} e^{\frac{1}{2} \pi^{ab} \iota_a \iota_b} \left[ \sum_{p : \text{even/odd}} \frac{1}{p!} \hat{\mathcal{F}}_{a_1 \cdots a_p} r^{a_1} \wedge \cdots \wedge r^{a_p} \right].
\tag{3.82}
$$

Note that here we are using the field strength in the A-basis (which satisfies $dF = 0$) and this is related to the one in C-basis as

$$
G = e^{-B_2 \wedge} F \,,
\tag{3.83}
$$

that satisfies the standard Bianchi identity

$$
dG + H_3 \wedge G = 0 \,,
\tag{3.84}
$$

when $G$ is independent of the dual coordinates $\tilde{x}_m$.

## 3.6  An example with Ramond–Ramond fluxes

Let us consider a 20-dimensional $DD^+$ with the structure constants

$$
f_{12}{}^2 = -1 \,, \quad f_{12}{}^3 = -1 \,, \quad f_{13}{}^2 = -1 \,, \quad f_{13}{}^3 = -1 \,, \quad Z_1 = -2 \,.
\tag{3.85}
$$

The non-trivial subalgebra generated by $\{T_1, T_2, T_3\}$ are known as $((\text{III}, -4\tilde{X}^1), (\text{I}, 0))$. Using the parameterization $g = e^{x T_1} e^{y T_2} e^{z T_3} e^{w_4 T_4} \cdots e^{w_{10} T_{10}}$, the non-trivial part of $v_a^m$ and $e_a^m$ are found as (the other components are just $v_a = e_a = \partial_a$)

$$
\begin{aligned}
&v_1 = \partial_x + (y + z)(\partial_y + \partial_z), \quad v_2 = \partial_y, \quad v_3 = \partial_z, \\
&e_1 = \partial_x, \quad e_2 = \tfrac{1}{2}\left[(e^{2x} + 1)\partial_y + (e^{2x} - 1)\partial_z\right], \quad e_3 = \tfrac{1}{2}\left[(e^{2x} - 1)\partial_y + (e^{2x} + 1)\partial_z\right].
\end{aligned}
\tag{3.86}
$$

We introduce constants

$$
\hat{g}_{ab} = \begin{pmatrix}
1 & 1 & 0 & 0 & & 0 \\
1 & 1 & 1 & 0 & & 0 \\
0 & 1 & 1 & 0 & & 0 \\
0 & 0 & 0 & 1 & & 0 \\
& & & & \ddots & \\
0 & 0 & 0 & 0 & & 1
\end{pmatrix}, \quad \hat{\beta}^{ab} = 0, \quad |\hat{\mathcal{F}}\rangle = 6\sqrt{2}\,\Gamma^1 \left[(\Gamma^2 + \Gamma^3)\Gamma^{4\cdots10} - 1\right]|0\rangle,
\tag{3.87}
$$

and then, by using $\Delta = -2x$ and supposing $\varphi = 0$, the supergravity fields are found as

$$
\begin{aligned}
&ds^2 = e^{4x}\left[dx^2 + dx\,(dy - dz) + ds_{T^7}^2\right] + e^{2x}\,dx\,(dy + dz) + (dy + dz)^2 \,, \\
&B_2 = 0, \qquad e^{-2\Phi} = e^{-16x}, \qquad F_1 = -6\sqrt{2}\,e^{-8x}\,dx \,,
\end{aligned}
\tag{3.88}
$$

where $ds_{T^7}^2 \equiv dw_4^2 + \cdots + dw_{10}^2$ is a seven-dimensional flat metric. This is a solution of type IIB* supergravity.

Now we consider a generalized Yang–Baxter deformation with

$$
r^{23} = \frac{\eta}{2} \,.
\tag{3.89}
$$

The resulting DD$^+$ has the structure constants

$$f_{12}{}^2 = -1, \quad f_{12}{}^3 = -1, \quad f_{13}{}^2 = -1, \quad f_{13}{}^3 = -1, \quad f_1{}^{23} = \eta, \quad Z_1 = -2. \tag{3.90}$$

The structure constants $f_1{}^{23}$ produces the Jacobi–Lie structure $\pi = \frac{\eta}{2}(1 - e^{-2x})\partial_y \wedge \partial_u$ and the supergravity fields are

$$\begin{aligned}
ds^2 &= e^{4x}\big[dx^2 + dx(dy - dz) + ds_{T^7}^2\big] + e^{2x}dx(dy + dz) + (dy + dz)^2 - \tfrac{\eta^2}{4}e^{4x}dx^2, \\
B_2 &= -\tfrac{\eta}{2}e^{2x}dx \wedge (dy + dz), \qquad e^{-2\Phi} = e^{-16x}, \qquad F_1 = -6\sqrt{2}\,e^{-8x}dx.
\end{aligned} \tag{3.91}$$

This is again a solution of type IIB$^*$ supergravity and the Jacobi–Lie $T$-duality indeed works as a solution generating technique.

## 3.7 Jacobi–Lie $T$-plurality with spectator fields

The inclusion of the spectator fields is straightforward similar to the case of the Poisson–Lie $T$-duality/$T$-plurality (see Appendix B of [22]). Here, instead of repeating the presentation of [22], we only comment on some non-trivialities that are specific to the Jacobi–Lie $T$-plurality.

We consider a ten-dimensional spacetime with the "internal coordinates" $x^m$ ($m = 1, \ldots, D$) and the "external coordinates" $y^\mu$ ($\mu = D + 1, \ldots, 10$). In the string sigma model, the scalar fields $y^\mu(\sigma)$ are called the spectator fields because they are invariant under the non-Abelian duality. We formally double all of the directions, and the generalized coordinates are given by $x^M = (x^m, \tilde{x}_m, y^\mu, \tilde{y}_\mu)$. The "flat" indices $A, B$ and $\mathcal{A}, \mathcal{B}$ also run over the 20 directions. The underlying algebra DD$^+$ is associated with the $2D$-dimensional doubled coordinates $\{x^m, \tilde{x}_m\}$, and for example, the generalized frame fields constructed in the previous sections are embedded into the first $2D \times 2D$-block of the $20 \times 20$ matrix $\mathcal{E}_A{}^M$. We assume that $\mathcal{E}_A{}^M$ and the double vielbein $V_A{}^{\mathcal{B}}$ have block-diagonal forms, i.e., they are given by direct sums of the $2D \times 2D$-block associated with the internal directions and $(20 - 2D) \times (20 - 2D)$-blocks associated with the external directions. In particular, we suppose that the external block of $\mathcal{E}_A{}^M$ is an identity matrix. With such understanding, the conditions for the Jacobi–Lie symmetry in the presence of the spectator fields, but without the R–R fields, are given by

$$\mathcal{H}_{MN} = \mathcal{E}_M{}^A(x)\,\mathcal{E}_N{}^B(x)\,\hat{\mathcal{H}}_{AB}(y), \qquad \hat{\mathcal{H}}_{AB}(y) \equiv V_A{}^{\mathcal{A}}(y)\,V_B{}^{\mathcal{B}}(y)\,\hat{\mathcal{H}}_{\mathcal{AB}}, \tag{3.92}$$

$$e^{-2d} = e^{-2\hat{d}(y)}\,e^{-2d(x)}, \qquad e^{-2d(x)} \equiv e^{-2\varphi(x)}\,e^{-\Delta}\,\tilde{\sigma}^{D-\frac{3}{2}}\,|\det \ell_m^a|. \tag{3.93}$$

The difference is that $V_A{}^{\mathcal{A}}(y)$ is no longer constant and that the dilaton also acquires the $y$-dependence $\hat{d}(y)$. By following the same discussion as [22], we can show that the O$(D, D)$ transformation which rotates the internal indices is a symmetry of the equation of motion.

When the R–R fields are also present, the symmetry becomes slightly subtle. In the presence of the spectator fields, the tensor $\mathcal{G}^{\mathcal{AB}}$ becomes

$$\begin{aligned}
\mathcal{G}^{\mathcal{AB}} \equiv\ & 2\,\hat{\mathcal{H}}^{\mathcal{C}[\mathcal{A}}\mathcal{D}^{\mathcal{B}]}\mathcal{F}_{\mathcal{C}} - \tfrac{1}{2}\hat{\mathcal{H}}^{\mathcal{CD}}\big(\eta^{\mathcal{AE}}\eta^{\mathcal{BF}} - \hat{\mathcal{H}}^{\mathcal{AE}}\hat{\mathcal{H}}^{\mathcal{BF}}\big)\big(\mathcal{F}_{\mathcal{C}} - \mathcal{D}_{\mathcal{C}}\big)\mathcal{F}_{\mathcal{DEF}} \\
& - \hat{\mathcal{H}}_{\mathcal{D}}{}^{[\mathcal{A}}\big(\mathcal{F}_{\mathcal{C}} - \mathcal{D}_{\mathcal{C}}\big)\mathcal{F}^{\mathcal{B}]\mathcal{CD}} + \tfrac{1}{2}\big(\eta^{\mathcal{CE}}\eta^{\mathcal{DF}} - \hat{\mathcal{H}}^{\mathcal{CE}}\hat{\mathcal{H}}^{\mathcal{DF}}\big)\hat{\mathcal{H}}^{\mathcal{G}[\mathcal{A}}\mathcal{F}_{\mathcal{CD}}{}^{\mathcal{B}]}\mathcal{F}_{\mathcal{EFG}},
\end{aligned} \tag{3.94}$$

where $\mathcal{D}_{\mathcal{A}} \equiv V_{\mathcal{A}}{}^B\,\mathcal{E}_B{}^M\,\partial_M$ and the fluxes contain both the external and internal parts:

$$\mathcal{F}_{\mathcal{A}} = \hat{\mathcal{F}}(y) + e^{\omega(x)}V_{\mathcal{A}}{}^B(y)\mathcal{F}_B, \tag{3.95}$$

$$\mathcal{F}_{\mathcal{ABC}} = \hat{\mathcal{F}}_{\mathcal{ABC}}(y) + e^{\omega(x)}V_{\mathcal{A}}{}^D(y)V_{\mathcal{B}}{}^E(y)V_{\mathcal{C}}{}^F(y)F_{DEF}. \tag{3.96}$$

The internal/external parts contribute to the internal/external components of the matrix $\mathcal{G}^{\mathcal{AB}}$, respectively. Then, the internal components of $\mathcal{G}^{\mathcal{AB}}$ (or $\mathcal{S}_{MN}$) scale as $e^{2\omega}$ while the external

components are independent of $\omega$. In order to realize the equations of motion $\mathcal{S}_{MN} = \mathcal{E}_{MN}$, the energy-momentum tensor $\mathcal{E}_{MN}$ also should scale in the same way, but it is non-trivial.

Then we can consider two possibilities: $(i)$ the external components of $\mathcal{S}_{MN}$ vanish, or $(ii)$ the internal components of $\mathcal{S}_{MN}$ vanish by themselves. The former is the case studied in the previous sections. In that case, we choose the R–R fields as

$$|F\rangle = \sqrt{\det(e^\omega e_a^m)}\, e^{-\mathsf{d}(x)}\, e^{\omega(x)}\, S_U|\hat{\mathcal{F}}(y)\rangle\,, \tag{3.97}$$

which is a natural extension of (3.77) including the $y$-dependence into $|\hat{\mathcal{F}}\rangle$. In the latter case, the scale factor $e^{\omega(x)}$ is not necessary and we consider

$$|F\rangle = \sqrt{\det(e^\omega e_a^m)}\, e^{-\mathsf{d}(x)}\, S_U|\hat{\mathcal{F}}(y)\rangle\,. \tag{3.98}$$

In terms of the differential form, this can be expressed as

$$F = e^{-\varphi(x)}\, e^{\frac{D-1}{2}\Delta}\, \tilde{\sigma}^{\frac{D-3}{4}} |\det a_a{}^b|^{\frac{1}{2}}\, e^{\frac{1}{2}\pi^{ab}\iota_a\iota_b} \left[ \sum_{p:\text{even/odd}} \frac{1}{p!}\, \hat{\mathcal{F}}_{\hat{a}_1\cdots\hat{a}_p}(y)\, \mathcal{E}^{\hat{a}_1} \wedge \cdots \wedge \mathcal{E}^{\hat{a}_p} \right], \tag{3.99}$$

where we have defined $\mathcal{E}^{\hat{a}} \equiv \mathcal{E}^{\hat{a}}{}_{\hat{m}}\, dx^{\hat{m}}$ with $x^{\hat{m}} \equiv (x^m, y^\mu)$ and $\{\hat{a}\} = \{a, \dot{\mu}\}$. Here, the dotted indices $\{\dot{\mu}\}$ denote the "flat" indices associated with $\{\mu\}$ and $\mathcal{E}^{\hat{a}}{}_{\hat{m}}$ is a component of $\mathcal{E}_A{}^M$.

The existence of the two options are specific to the Jacobi–Lie $T$-plurality, and these two are degenerate in the case of the Poisson–Lie $T$-duality (where $\omega = 0$). In the next subsection, we present an example using the latter option (3.98).

## 3.8 An example with spectator fields

We consider an eight-dimensional $DD^+$ $(D = 4)$ with the structure constants given in Eq. (3.85). We introduce the ten-dimensional coordinates

$$\{x^m; y^\mu\} = \{x, y, u, v; z, r, \xi, \phi_1, \phi_2, \phi_3\}\,, \tag{3.100}$$

and $y^\mu$ are the spectator fields. Using the parameterization $g = e^{x\,T_1}\, e^{y\,T_2}\, e^{u\,T_3}\, e^{v\,T_4}$, we obtain the left-/right-invariant vector fields as given in Eq. (3.86). We choose the metric $\hat{g}_{ab}(y)$, dilaton $\hat{d}(y)$, the R–R field $|\hat{\mathcal{F}}(y)\rangle$, and $\varphi(x)$ as

$$\hat{g}_{ab} = \begin{pmatrix} \begin{array}{ccccc} \frac{1}{z^2} & \frac{1}{z^2} & 0 & 0 & 0 \\ \frac{1}{z^2} & \frac{1}{z^2} & -\frac{1}{z^2} & 0 & 0 \\ 0 & -\frac{1}{z^2} & \frac{1}{z^2} & 0 & 0 \\ 0 & 0 & 0 & \frac{1}{z^2} & 0 \\ 0 & 0 & 0 & 0 & \frac{1}{z^2} \end{array} & 0_{5\times5} \\ \hline 0_{5\times5} & g_{S^5} \end{pmatrix}, \quad \hat{\beta}^{ab} = 0\,, \quad e^{-2\hat{d}} = \frac{\cos r \cos\xi \sin^3 r \sin\xi}{z^5}\,, \tag{3.101}$$

$$|\hat{\mathcal{F}}\rangle = 4\left(-z^{-5}\,\Gamma^{\dot{u}\dot{v}\dot{x}\dot{y}\dot{z}} + \sin^3 r \cos r \sin\xi \cos\xi\,\Gamma^{\dot{r}\dot{\xi}\dot{\phi}_1\dot{\phi}_2\dot{\phi}_3}\right)\,, \qquad \varphi = -2x\,,$$

where the metric $g_{S^5}$ on $S^5$ corresponds to the line element

$$ds_{S^5}^2 \equiv dr^2 + \sin^2 r\left(d\xi^2 + \cos^2\xi\, d\phi_1^2 + \sin^2\xi\, d\phi_2^2\right) + \cos^2 r\, d\phi_3^2\,. \tag{3.102}$$

Using $\pi^{ab} = 0$ and $\Delta = -2x$, the generalized frame fields become

$$\mathcal{E}_A{}^M(x) = \begin{pmatrix} \begin{array}{cccccccc} e^{-2x} & 0 & 0 & 0 & 0 & 0 & 0 & 0 \\ 0 & e^{-x}\cosh x & e^{-x}\sinh x & 0 & 0 & 0 & 0 & 0 \\ 0 & e^{-x}\sinh x & e^{-x}\cosh x & 0 & 0 & 0 & 0 & 0 \\ 0 & 0 & 0 & e^{-2x} & 0 & 0 & 0 & 0 \\ 0 & 0 & 0 & 0 & e^{2x} & 0 & 0 & 0 \\ 0 & 0 & 0 & 0 & 0 & e^x\cosh x & -e^x\sinh x & 0 \\ 0 & 0 & 0 & 0 & 0 & -e^x\sinh x & e^x\cosh x & 0 \\ 0 & 0 & 0 & 0 & 0 & 0 & 0 & e^{2x} \end{array} & 0 \\ \hline 0 & 1_{12\times12} \end{pmatrix}. \tag{3.103}$$

By acting the twist, we find that this is the AdS$_5\times$S$^5$ solution of type IIB supergravity,

$$ds^2_{AdS_5\times S^5} = z^{-2}\left(ds^2_{4D} + dz^2\right) + ds^2_{S^5}, \qquad B_2 = 0, \qquad \Phi = 0,$$
$$ds^2_{4D} \equiv e^{4x}\left[dx^2 + dx\,dy + dy^2 + du^2 - du\,(dx + 2\,dy) + dv^2\right] + e^{2x}\,dx\,(du + dy), \quad (3.104)$$
$$F = 4\left[-\frac{e^{6x}\,dx \wedge dy \wedge du \wedge dv \wedge dz}{z^5} + \sin^3 r\cos r\sin\xi\cos\xi\,dr \wedge d\xi \wedge d\phi_1 \wedge d\phi_2 \wedge d\phi_3\right].$$

Here we have used $e^{-\varphi(x)}\,e^{\frac{D-1}{2}\omega(x)}|\det a_a{}^b|^{\frac{1}{2}} = 1$ (where $D = 4$), and

$$\mathcal{E}^1 \wedge \cdots \wedge \mathcal{E}^4 \wedge \mathcal{E}^{\dot z} = e^{6x}\,dx \wedge dy \wedge du \wedge dv \wedge dz. \tag{3.105}$$

Again we perform a generalized Yang–Baxter deformation (3.89) and obtain the DD$^+$ given in Eq. (3.90). The $\omega$ is not changed and the Jacobi–Lie structure is $\pi = \frac{\eta}{2}\left(1 - e^{-2x}\right)\partial_y \wedge \partial_u$. The deformed geometry is

$$ds^2 = ds^2_{AdS_5\times S^5} - \frac{\eta^2\,e^{4x}(2\,e^{2x} - 1)^2 dx^2}{4\,z^6}, \qquad B_2 = \frac{\eta\left(e^{6x} - \frac{1}{2}\,e^{4x}\right)}{z^4}\,dx \wedge (dy - du),$$
$$\Phi = 0, \qquad G_3 = \frac{2\,\eta\,e^{5x}\,(\cosh x + 3\sinh x)\,dx \wedge dv \wedge dz}{z^5}, \tag{3.106}$$
$$G_5 = 4\left[-\frac{e^{6x}\,dx \wedge dy \wedge du \wedge dv \wedge dz}{z^5} + \sin^3 r\cos r\sin\xi\cos\xi\,dr \wedge d\xi \wedge d\phi_1 \wedge d\phi_2 \wedge d\phi_3\right].$$

This also satisfies the type IIB supergravity equations of motion.

In order to perform more interesting Jacobi–Lie $T$-plurality, the classification of the six-dimensional DD$^+$ will be useful. The classification of the Jacobi–Lie bialgebra has been done in [25] but which bialgebras are in the same orbit O($D, D$) rotations have not been studied. If such a classification is worked out, we may find more dual geometries from the AdS$_5\times$S$^5$ solution (3.104).

# 4 Jacobi–Lie $T$-plurality in string theory

In the string sigma model, we can clearly see the symmetry of the Poisson–Lie $T$-duality by using a formulation called the $\mathcal{E}$-model [38]. The $\mathcal{E}$-model is defined by a Hamiltonian

$$H = \frac{1}{4\pi\alpha'}\int d\sigma\,\hat{\mathcal{H}}^{AB}\,j_A(\sigma)\,j_B(\sigma), \tag{4.1}$$

and the current algebra

$$\{j_A(\sigma), j_B(\sigma')\} = F_{AB}{}^C\,j_C(\sigma) + \eta_{AB}\,\delta'(\sigma - \sigma'), \tag{4.2}$$

where $\hat{\mathcal{H}}^{AB}$ is a constant O($D, D$) matrix and $F_{AB}{}^C$ are certain structure constants. The dynamics is governed by the O($D, D$)-manifest equations (4.1) and (4.2), and the time evolution of the currents can be determined by

$$\partial_\tau j_A = \{j_A, H\}. \tag{4.3}$$

In fact, Eq. (4.3) is exactly the equations of motion of string theory defined on a target space with the generalized metric

$$\mathcal{H}_{MN} = E_M{}^A E_N{}^B\,\hat{\mathcal{H}}_{AB}, \tag{4.4}$$

where $E_A{}^M$ are the generalized frame fields satisfying $\hat{\mathcal{L}}_{E_A} E_B = -F_{AB}{}^C E_C$ with $F_{AB}{}^C$ the structure constants of a Drinfel'd double. Here, the currents have been identified as

$$j_A(\sigma) = E_A{}^M(x(\sigma)) Z_M(\sigma), \qquad Z_M(\sigma) \equiv \begin{pmatrix} p_m(\sigma) \\ \partial_\sigma x^m(\sigma) \end{pmatrix}, \tag{4.5}$$

where $p_m$ are the canonical momenta associated with $x^m$. The current algebra (4.2) is simply a rewriting of the canonical commutation relation

$$\{Z_M(\sigma), Z_N(\sigma')\} = \eta_{MN}\, \delta'(\sigma - \sigma'), \tag{4.6}$$

by using $\hat{\mathcal{L}}_{E_A} E_B = -F_{AB}{}^C E_C$. Under the Poisson–Lie $T$-duality/$T$-plurality $T_A \to C_A{}^B T_B$, we get a new generalized frame fields $E_A'{}^M$ satisfying $\hat{\mathcal{L}}_{E_A'} E_B' = -F_{AB}'{}^C E_C'$ (where $F_{AB}'{}^C \equiv C_A{}^D C_B{}^E (C^{-1})_F{}^C F_{DE}{}^F$) and we define the dual currents as $j_A'(\sigma) = E_A'{}^M(x(\sigma)) Z_M'(\sigma)$ where $Z_M'$ satisfies the canonical commutation relation (4.6). The Hamiltonian for string theory on the dual geometry can be expressed as

$$H' = \frac{1}{4\pi\alpha'} \int d\sigma\, \hat{\mathcal{H}}'^{AB}\, j_A'(\sigma)\, j_B'(\sigma), \tag{4.7}$$

where $\hat{\mathcal{H}}'^{AB} \equiv (C^{-1})_C{}^A (C^{-1})_D{}^B \hat{\mathcal{H}}^{CD}$ and the dual currents satisfies the algebra

$$\{j_A'(\sigma), j_B'(\sigma')\} = F_{AB}'{}^C j_C(\sigma) + \eta_{AB}\, \delta'(\sigma - \sigma'). \tag{4.8}$$

Then the currents $j_A'$ follow the same time evolution as $C_A{}^B j_B$, and we can clearly see the covariance of the string equations of motion. In [27], the currents $j_A$ was regarded as the phase-space variables and the Poisson–Lie $T$-duality/$T$-plurality $j_A \to j_A'$ that preserves the Hamiltonian was regarded as a canonical transformation.

Now let us consider the case of the Jacobi–Lie $T$-plurality. Again the generalized metric is expressed as

$$\mathcal{H}_{MN} = \mathcal{E}_M{}^A \mathcal{E}_N{}^B \hat{\mathcal{H}}_{AB}, \tag{4.9}$$

where $\mathcal{E}_M{}^A$ satisfies

$$\hat{\mathcal{L}}_{\mathcal{E}_A} \mathcal{E}_B{}^M = -e^\omega \big(X_{AB}{}^C - 2 Z_{[A} \delta_{B]}^C - \eta_{AB} Z^C\big) \mathcal{E}_C{}^M = -e^\omega F_{AB}{}^C \mathcal{E}_C{}^M, \tag{4.10}$$

and $F_{AB}{}^C$ is the one given in (2.5). Then introducing the currents

$$\mathcal{J}_A(\sigma) \equiv \mathcal{E}_A{}^M(x(\sigma)) Z_M(\sigma), \tag{4.11}$$

we obtain the Hamiltonian and the current algebra as

$$H = \frac{1}{4\pi\alpha'} \int d\sigma\, \hat{\mathcal{H}}^{AB}\, \mathcal{J}_A(\sigma)\, \mathcal{J}_B(\sigma), \tag{4.12}$$

$$\{\mathcal{J}_A(\sigma), \mathcal{J}_B(\sigma')\} = e^{\omega(x(\sigma))} F_{AB}{}^C \mathcal{J}_C(\sigma) + \eta_{AB}\, \delta(\sigma - \sigma'). \tag{4.13}$$

Due to the appearance of the explicit $x$-dependence in $e^{\omega(x(\sigma))}$, we cannot treat the currents as the phase-space variables, but as complicated functions of $x(\sigma)$ and their canonical conjugate momenta. Then the Hamiltonian also needs to be regarded as a non-linear function. Consequently, the covariance under the Jacobi–Lie $T$-plurality is not manifest.

Let us also discuss the covariance from another perspective. If we start with the action

$$S = -\frac{1}{4\pi\alpha'} \int_\Sigma d^2\sigma\, \sqrt{-\gamma}\, \big(\gamma^{\alpha\beta} - \varepsilon^{\alpha\beta}\big) \big(g_{mn} + B_{mn}\big) \partial_\alpha x^m\, \partial_\beta x^n, \tag{4.14}$$

the equations of motion can be expressed as

$$dJ_a = \frac{1}{2}\left(£_{v_a} g_{mn}\, dx^m \wedge *dx^n + £_{v_a} B_{mn}\, dx^m \wedge dx^n\right),\tag{4.15}$$

where

$$J_a \equiv v_a^m\left(g_{mn} * dx^n + B_{mn}\, dx^n\right).\tag{4.16}$$

If we identify the metric and the $B$-field as $g_{mn}+B_{mn} = E_{mn}$, by using Eq. (3.10), the equations of motion can be rewritten in a suggestive form [17]

$$dJ_a = \tfrac{1}{2}\, e^{-2\Delta}(f_a{}^{bc} + 2\,\delta_a^b Z^c - 2\,\delta_a^c Z^b)J_b \wedge J_c.\tag{4.17}$$

However we cannot say anything more from this relation.

In the case of the Poisson–Lie $T$-duality, where $\Delta = 0$ and $Z^a = 0$, we can regard the relation (4.17) as a Maurer–Cartan equation and identify the current $J_a$ as the right-invariant 1-form

$$d\tilde{g}\,\tilde{g}^{-1} = J_a\, T^a,\qquad \tilde{g} \equiv e^{\tilde{x}_a T^a}.\tag{4.18}$$

Then, we can rewrite the equations of motion in a manifestly $O(D,D)$-covariant form as (see section 6.1 of [22] for the details)

$$\hat{\mathcal{P}}^A = \hat{\mathcal{H}}^A{}_B * \hat{\mathcal{P}}^B,\tag{4.19}$$

where $\hat{\mathcal{P}}^A$ is constructed by using an element of the Drinfel'd double $l \equiv g\,\tilde{g}$ as

$$\hat{\mathcal{P}} \equiv \hat{\mathcal{P}}^A T_A \equiv dl\, l^{-1}.\tag{4.20}$$

The equations of motion can be also expressed as the $O(D,D)$ covariant Maurer–Cartan equation for the Drinfel'd double

$$d\hat{\mathcal{P}}^A + \frac{1}{2}\, F_{BC}{}^A \hat{\mathcal{P}}^B \wedge \hat{\mathcal{P}}^C = 0.\tag{4.21}$$

In the case of the Jacobi–Lie $T$-duality of [17], due to the presence of $\Delta$ in Eq. (4.17), $J_a$ cannot be expressed by using $\tilde{g}$ and it is not clear how to construct a covariant or geometric object similar to $\hat{\mathcal{P}}^A$. If we instead identify the metric and the $B$-field as $g_{mn} + B_{mn} = \mathcal{E}_{mn}$ as in the case of the Jacobi–Lie $T$-plurality, Eq. (3.8) leads to

$$dJ_a = -\tfrac{\breve{\sigma}^{-1}}{2}\,(f_a{}^{bc} + 2\,\delta_a^b Z^c - 2\,\delta_a^c Z^b)J_b \wedge J_c - 2Z_b\, r^b \wedge J_a.\tag{4.22}$$

In this case, there is no scale factor, but due to the presence of the last term, this again cannot be regarded as a Maurer–Cartan equation. According to the above considerations, we suspect that the Jacobi–Lie $T$-plurality is not a symmetry of the string sigma model.

One of the reasons for the issue may be that the $DD^+$ is a Leibniz algebra instead of a Lie algebra. In the case of the Poisson–Lie $T$-duality, a string is fluctuating on the Drinfel'd double and the position of the string is described by a map, $l : \Sigma \to \mathcal{D}$, from the worldsheet to a Drinfel'd double $\mathcal{D}$. However, in the case of the Leibniz algebra, a group-like global structure is complicated and it is not clear how to describe the position of the string on the doubled geometry similar to the case of the Drinfel'd double. A recent study [39] may be useful in clarifying this point.

## 5 Conclusions

In this paper, we proposed a Leibniz algebra $DD^+$ and showed that this provides an alternative description of the Jacobi–Lie bialgebra. Extending the standard procedure developed in the Poisson–Lie $T$-duality, we showed that a $DD^+$ systematically constructs a Jacobi–Lie structures and the generalized frame fields satisfying $\hat{\mathcal{L}}_{E_A} E_B = -X_{AB}{}^C E_C$ . Using the generalized frame fields, we proposed a natural extension of the Poisson–Lie $T$-duality, which we call the Jacobi–Lie $T$-plurality. We then showed that the Jacobi–Lie $T$-plurality (with the R–R fields and the spectator fields) is a symmetry of the equations of motion of DFT. As a demonstration, we provided several examples of the Jacobi–Lie $T$-plurality. At the level of the string sigma model, we were faced with a difficulty in the realization of the Jacobi–Lie $T$-plurality, and this may indicate that the scale symmetry $\mathbb{R}^+$ is not a (classical) symmetry of string theory. To clarify the status of this scale symmetry, it is important to check whether the Jacobi–Lie $T$-plurality remains as a symmetry of $\alpha'$-corrected supergravity by extending recent works on the Poisson–Lie $T$-duality [40–42].

In M-theory, the exceptional Drinfel'd algebra (associated with the SL(5) duality group) has been found as

$$
\begin{aligned}
T_a \circ T_b &= f_{ab}{}^c\, T_c \,, \qquad T^{a_1 a_2} \circ T^{b_1 b_2} = -2 f_c{}^{a_1 a_2 [b_1}\, T^{b_2] c} \,, \\
T_a \circ T^{b_1 b_2} &= f_a{}^{b_1 b_2 c}\, T_c + 2 f_{ac}{}^{[b_1}\, T^{b_2] c} + 3 Z_a\, T^{b_1 b_2} \,, \\
T^{a_1 a_2} \circ T_b &= -f_b{}^{a_1 a_2 c}\, T_c + 3 f_{[c_1 c_2}{}^{[a_1}\, \delta_{b]}^{a_2]}\, T^{c_1 c_2} - 9 Z_c\, \delta_b^{[c}\, T^{a_1 a_2]} \,.
\end{aligned}
\tag{5.1}
$$

If we decompose the index as $a = \{\dot{a}, \sharp\}$ and assume $f_{\dot{a}b}{}^\sharp = 0$ , we find that the generators $\{T_{\dot{a}}, T^{\dot{a}} \equiv T^{\dot{a}\sharp}\}$ satisfy the subalgebra

$$
\begin{aligned}
T_{\dot{a}} \circ T_{\dot{b}} &= f_{\dot{a}\dot{b}}{}^{\dot{c}}\, T_{\dot{c}} \,, \qquad T^{\dot{a}} \circ T^{\dot{b}} = -f_{\dot{c}}{}^{\dot{a}\dot{b}\sharp}\, T^{\dot{c}} \,, \\
T_{\dot{a}} \circ T^{\dot{b}} &= -f_{\dot{a}}{}^{\dot{b}\dot{c}\sharp}\, T_{\dot{c}} - f_{\dot{a}\dot{c}}{}^{\dot{b}}\, T^{\dot{c}} + (3 Z_{\dot{a}} - f_{\dot{a}\sharp}{}^\sharp)\, T^{\dot{b}} \,, \\
T^{\dot{a}} \circ T_{\dot{b}} &= f_{\dot{b}}{}^{\dot{a}\dot{c}\sharp}\, T_{\dot{c}} + f_{\dot{b}\dot{c}}{}^{\dot{a}}\, T^{\dot{c}} - (3 Z_{\dot{b}} - f_{\dot{b}\sharp}{}^\sharp)\, T^{\dot{a}} + (3 Z_{\dot{c}} - f_{\dot{c}\sharp}{}^\sharp)\, \delta_{\dot{b}}^{\dot{a}}\, T^{\dot{c}} \,.
\end{aligned}
\tag{5.2}
$$

This is noting but the $DD^+$ under the identifications, $f_{\dot{a}}{}^{\dot{b}\dot{c}} = -f_{\dot{a}}{}^{\dot{b}\dot{c}\sharp}$, $Z^{\dot{a}} = 0$, and $2 Z_{\dot{a}} = 3 Z_{\dot{a}} - f_{\dot{a}\sharp}{}^\sharp$ . Similarly, the extended Drinfel'd algebra in the type IIB picture also contains the $DD^+$ as a subalgebra. Thus, the Jacobi–Lie $T$-plurality is a subset of the proposed Nambu–Lie $U$-duality.[6] An issue in the Nambu–Lie $U$-duality is that the equations of motion of the exceptional field theory are complicated and the covariance under the Nambu–Lie $U$-duality cannot be easily proven. The results of this paper show that the non-Abelian duality works as a solution generating transformation even when the $Z_A$ is present. Further steps towards the proof of Nambu–Lie $U$-duality will be taken in future work.

Another future direction is to study a $U$-duality extension of the Jacobi–Lie structure. For this purpose, we need to study the Nambu–Jacobi structure [45] on a group manifold. In this paper, we have constructed the Jacobi–Lie structure $\pi$ and $E$ from a $DD^+$, and the vector field $E \propto Z^a\, e_a$ is associated with the vector $Z_A = (Z_a, Z^a)$. In the case of the EDA (in the M-theory picture), $\pi$ is replaced by a tri-vector $\pi^{(3)}$ and $E$ will be replaced by a bi-vector $E^{(2)} \propto Z^{ab}\, e_a e_b$ because $Z_A$ is replaced by $Z_A = (Z_a, \frac{Z^{a_1 a_2}}{\sqrt{2!}})$. In the literature, the non-Abelian $U$-duality is studied by assuming $Z^{a_1 a_2} = 0$, but this assumption may not be necessary. It will be an interesting future work to keep $Z^{a_1 a_2}$ to find a generalized non-Abelian $U$-duality. It is also interesting to study the associated generalized Yang–Baxter deformation.

---

[6]We note that some $DD^+$ cannot be embedded into the extended Drinfel'd algebra (see [4,6]), and accordingly, some Jacobi–Lie $T$-plurality cannot be realized as a Nambu–Lie $U$-duality.

### Acknowledgments

We thank Kentaroh Yoshida for a helpful correspondence. We also thank the anonymous referee for the careful reading of the manuscript and remarks that helped us to remove an unnecessary restriction $Z^a = 0$ in the original discussion on the Jacobi–Lie $T$-plurality. The work of JJFM is supported by Universidad de Murcia-Plan Propio Postdoctoral, the Spanish Ministerio de Economía y Competitividad and CARM Fundación Séneca under grants FIS2015-28521 and 21257/PI/19. The work of YS is supported by JSPS Grant-in-Aids for Scientific Research (C) 18K13540 and (B) 18H01214.

## A  Embedding tensors in half-maximal 7D SUGRA and DD⁺

In this appendix, we conduct a detailed study of the relationship between the embedding tensors in half-maximal 7D gauged supergravity and the DD⁺. Among the 13 inequivalent orbits classified in [28], we show that orbits 2, 3, 5, 7,..., 13 can be mapped to some DD⁺s by performing O(3, 3) redefinitions of generators $T_A \rightarrow C_A{}^B T_B$. For each orbit, the matrix $C_A{}^B$ (which is not unique) is found by trial and error. For orbit 4 or 6, only when $\alpha = 0$, we find such a matrix $C_A{}^B$ but failed to find such matrix for $\alpha \neq 0$. For orbit 1, as we explain below, we conclude that this is not related to any DD⁺.

In the following, we use a short-hand notation,

$$c \equiv \cos\alpha, \qquad s \equiv \sin\alpha, \qquad t \equiv \tan\alpha. \tag{A.1}$$

Because of $-\frac{\pi}{4} < \alpha \leq \frac{\pi}{4}$, we have $-\frac{1}{\sqrt{2}} < s \leq c \leq \frac{1}{\sqrt{2}}$ and $1 < t \leq 1$. In addition, as was classified in [43, 44], there are 22 six-dimensional Drinfel'd doubles, which are called DD1,...,DD22, and we use the notation in the following. As we show in the following, all of these are related to some embedding tensors with $\xi_0 = 0$ (recall that a DD⁺ reduces to a Drinfel'd double when $\xi_0 = 0$).

### Orbit 1

Orbit 1 contains the non-vanishing fluxes

$$
\begin{aligned}
H_{123} &= c, & f_1{}^{23} &= c, & f_2{}^{13} &= -c, & f_3{}^{12} &= c, \\
R^{123} &= s, & f_{23}{}^1 &= s, & f_{13}{}^2 &= -s, & f_{12}{}^3 &= s.
\end{aligned}
\tag{A.2}
$$

The 6D Lie algebra $[T_A, T_B] = X_{AB}{}^C T_C$ has been identified as SO(4) for $\alpha \neq \frac{\pi}{4}$ or SO(3) (times three-dimensional Abelian algebra) for $\alpha = \frac{\pi}{4}$. According to the classification of six-dimensional Drinfel'd double [43], there is no Drinfel'd double whose Lie algebra is SO(4) or SO(3), and thus orbit 1 is not related to any Drinfel'd double.

### Orbit 2

Orbit 2 contains the non-vanishing fluxes

$$
\begin{aligned}
H_{123} &= c, & f_1{}^{23} &= c, & f_2{}^{13} &= -c, & f_3{}^{12} &= -c, \\
R^{123} &= s, & f_{23}{}^1 &= s, & f_{13}{}^2 &= -s, & f_{12}{}^3 &= -s.
\end{aligned}
\tag{A.3}
$$

Let us classify the range of parameter $\alpha$ into three categories.

1. $\underline{\alpha = 0}$   In this case, performing an O(3,3) transformation with

$$
C_A{}^B = \begin{pmatrix} 0 & 1 & 0 & 1 & 0 & 0 \\ -1 & 0 & 0 & 0 & 1 & 0 \\ 0 & 0 & 0 & 0 & 0 & 1 \\ \frac{1}{2} & 0 & 0 & 0 & \frac{1}{2} & 0 \\ 0 & \frac{1}{2} & 0 & -\frac{1}{2} & 0 & 0 \\ 0 & 0 & 1 & 0 & 0 & 0 \end{pmatrix},
\tag{A.4}
$$

we get a Drinfel'd double with

$$
f_{13}{}^2 = -1 , \quad f_{23}{}^1 = 1 , \quad f_1{}^{13} = 1 , \quad f_2{}^{23} = 1 .
\tag{A.5}
$$

According to [43], this corresponds to a Manin triple $(\mathbf{7_0}|\mathbf{5.ii}|b)$ with $b = 1$, which corresponds to the Drinfel'd double

$$
\text{DD1:} \qquad (\mathbf{9}|\mathbf{5}|b) \cong (\mathbf{8}|\mathbf{5.ii}|b) \cong (\mathbf{7_0}|\mathbf{5.ii}|b) \qquad (b > 0) .
\tag{A.6}
$$

2. $\underline{0 < \alpha}$   Here, performing an O(3,3) transformation with

$$
C_A{}^B = \begin{pmatrix} 0 & 0 & -\frac{1}{s} & 0 & 0 & 0 \\ 0 & \frac{1}{2} & 0 & -\frac{1}{2} & 0 & 0 \\ \frac{1}{2} & 0 & 0 & 0 & \frac{1}{2} & 0 \\ 0 & 0 & 0 & 0 & 0 & -s \\ -1 & 0 & 0 & 0 & 1 & 0 \\ 0 & 1 & 0 & 1 & 0 & 0 \end{pmatrix},
\tag{A.7}
$$

we get a Drinfel'd double with

$$
\begin{aligned}
f_{12}{}^2 &= -\frac{1}{t} , \quad f_{12}{}^3 = 1 , \quad f_{13}{}^2 = -1 , \quad f_{13}{}^3 = -\frac{1}{t} , \\
f_2{}^{12} &= -s^2 , \quad f_2{}^{13} = -cs , \quad f_3{}^{12} = cs , \quad f_3{}^{13} = -s^2 .
\end{aligned}
\tag{A.8}
$$

This corresponds to a Manin triple $(\mathbf{7_a}|\mathbf{7_{1/a}}|b)$ with $\mathbf{a} \equiv \frac{1}{t}$ and $b = cs = \frac{\mathbf{a}}{1+\mathbf{a}^2}$. This corresponds to the Drinfel'd double

$$
\text{DD3:} \qquad (\mathbf{7_a}|\mathbf{7_{1/a}}|b) \cong (\mathbf{7_{1/a}}|\mathbf{7_a}|b) \qquad (\mathbf{a} \geq 1, \ b \neq 0) .
\tag{A.9}
$$

3. $\underline{\alpha < 0}$   This case is related to the previous case through $T_1 \to -T_1$ and $T_2 \leftrightarrow T_3$.

As one can see from this example, each orbit of [28] corresponds to several different Drinfel'd doubles, each of which has several different decompositions into Manin triples. We also note that the Lie algebra of the two Drinfel'd doubles, DD1 and DD3, are isomorphic to $\text{SO}(3,1) \cong \text{SL}(2) \times \text{SL}(2)$ (see the first paragraph of section 4.1 in [43] for more details). The difference between DD1 and DD3 is in the definition of the bilinear form $\langle T_A, T_B \rangle$ on the Lie algebra of $\text{SO}(3,1)$.

## Orbit 3

Orbit 3 contains the non-vanishing fluxes

$$
\begin{aligned}
H_{123} &= c , \quad f_1{}^{23} = c , \quad f_2{}^{13} = c , \quad f_3{}^{12} = -c , \\
R^{123} &= s , \quad f_{23}{}^1 = s , \quad f_{13}{}^2 = s , \quad f_{12}{}^3 = -s .
\end{aligned}
\tag{A.10}
$$

Again we consider three cases.

1. $\underline{\alpha = 0}$   Performing an O(3, 3) transformation with

$$
C_A{}^B = \begin{pmatrix} 0 & -1 & 0 & -1 & 0 & 0 \\ 1 & 0 & 0 & 0 & -1 & 0 \\ 0 & 0 & 0 & 0 & 0 & 1 \\ -\frac{1}{2} & 0 & 0 & 0 & -\frac{1}{2} & 0 \\ 0 & -\frac{1}{2} & 0 & \frac{1}{2} & 0 & 0 \\ 0 & 0 & 1 & 0 & 0 & 0 \end{pmatrix},
\tag{A.11}
$$

we get a Drinfel'd double with

$$
f_{13}{}^2 = 1, \quad f_{23}{}^1 = 1, \quad f_1{}^{13} = -1, \quad f_2{}^{23} = -1.
\tag{A.12}
$$

This corresponds to a Manin triple $(\mathbf{6_0}|\mathbf{5.iii}|b)$ with $b = 1$, which is contained in

$$
\text{DD2:} \qquad (\mathbf{8}|\mathbf{5.i}|b) \cong (\mathbf{6_0}|\mathbf{5.iii}|b) \qquad (b > 0).
\tag{A.13}
$$

The Lie algebra of this Drinfel'd double is isomorphic to SO(2, 2).

2. $\underline{0 < |\alpha| < \frac{\pi}{4}}$   Performing an O(3, 3) transformation with

$$
C_A{}^B = \begin{pmatrix} 0 & 0 & -\frac{1}{s} & 0 & 0 & 0 \\ 0 & \frac{1}{2} & 0 & -\frac{1}{2} & 0 & 0 \\ -\frac{1}{2} & 0 & 0 & 0 & -\frac{1}{2} & 0 \\ 0 & 0 & 0 & 0 & 0 & -s \\ -1 & 0 & 0 & 0 & 1 & 0 \\ 0 & -1 & 0 & -1 & 0 & 0 \end{pmatrix},
\tag{A.14}
$$

we get a Drinfel'd double with

$$
\begin{aligned}
f_{12}{}^2 &= -\frac{1}{t}, & f_{12}{}^3 &= -1, & f_{13}{}^2 &= -1, & f_{13}{}^3 &= -\frac{1}{t}, \\
f_2{}^{12} &= -s^2, & f_2{}^{13} &= -cs, & f_3{}^{12} &= -cs, & f_3{}^{13} &= -s^2.
\end{aligned}
\tag{A.15}
$$

This is a Manin triple $(\mathbf{6_a}|\mathbf{6_{1/a}.i}|b)$ with $\mathbf{a} \equiv \frac{1}{t}$ and $b = cs = \frac{\mathbf{a}}{1+\mathbf{a}^2}$. This corresponds to the Drinfel'd double

$$
\text{DD4:} \qquad (\mathbf{6_a}|\mathbf{6_{1/a}.i}|b) \cong (\mathbf{6_{1/a}.i}|\mathbf{6_a}|b) \qquad (\mathbf{a} > 1, \ b \neq 0).
\tag{A.16}
$$

The Lie algebra of this Drinfel'd double is also isomorphic to SO(2, 2) although the bilinear form is defined differently from DD2.

3. $\underline{\alpha = \frac{\pi}{4}}$   Substituting $\alpha = \frac{\pi}{4}$ to (A.15), we obtain a Manin triple $(\mathbf{3}|\mathbf{3.i}|b)$ with $b = \frac{1}{2}$. This corresponds to the Drinfel'd double

$$
\text{DD8:} \qquad (\mathbf{3}|\mathbf{3.i}|b) \qquad (b \neq 0),
\tag{A.17}
$$

whose Lie algebra is isomorphic to SO(2, 1).

## Orbit 4

Orbit 4 contains the non-vanishing fluxes

$$
H_{123} = c, \quad f_{12}{}^3 = s, \quad f_1{}^{23} = c, \quad f_2{}^{13} = -c.
\tag{A.18}
$$

Performing an O(3, 3) transformation with

$$C_A{}^B = \begin{pmatrix} \frac{1}{c} & 0 & 0 & 0 & 0 & 0 \\ 0 & 1 & 0 & 0 & 0 & 0 \\ 0 & 0 & 0 & 0 & 0 & 1 \\ 0 & 0 & 0 & c & 0 & 0 \\ 0 & 0 & 0 & 0 & 1 & 0 \\ 0 & 0 & 1 & 0 & 0 & 0 \end{pmatrix}, \tag{A.19}$$

this is mapped to a flux configuration

$$f_{12}{}^3 = 1, \quad f_{23}{}^1 = c^2, \quad f_{13}{}^2 = -1, \quad H_{123} = t. \tag{A.20}$$

For a general $\alpha$, due to the presence of $H_{123}$, this does not correspond to a Lie algebra of a Drinfel'd double. Let us consider two cases: $\alpha = 0$ and $\alpha \neq 0$.

1. $\underline{\alpha = 0}$    Only in this case, we get the Manin triple $(\mathbf{9}|\mathbf{1})$, which corresponds to the Drinfel'd double

$$\text{DD5:} \qquad (\mathbf{9}|\mathbf{1}). \tag{A.21}$$

   The Lie algebra of DD5 is isomorphic to ISO(3) $\cong$ CSO(3, 0, 1) [28].

2. $\underline{\alpha \neq 0}$    Here we considered a general O(3, 3) matrix of the form $\begin{pmatrix} \alpha & 0 \\ 0 & (a^{-1})^t \end{pmatrix}\begin{pmatrix} 1 & \beta \\ 0 & 1 \end{pmatrix}\begin{pmatrix} 1 & 0 \\ \gamma & 1 \end{pmatrix}$ with $\det \alpha \neq 0$, $\beta^t = -\beta$, and $\gamma^t = -\gamma$, and tried to realize $F_{abc} = F^{abc} = 0$. However, there is no real solution. If instead we perform a redefinition with

$$C_A'{}^B = \begin{pmatrix} 1 & 0 & 0 & -\frac{t}{c^2} & 0 & 0 \\ 0 & \frac{1}{c} & 0 & 0 & -\frac{t}{c} & 0 \\ 0 & 0 & \frac{1}{c} & 0 & 0 & -\frac{t}{c} \\ 0 & 0 & 0 & 1 & 0 & 0 \\ 0 & 0 & 0 & 0 & c & 0 \\ 0 & 0 & 0 & 0 & 0 & c \end{pmatrix} \notin O(3, 3), \tag{A.22}$$

   Eq. (A.20) becomes the same algebra as $\alpha = 0$ (i.e., $f_{12}{}^3 = 1$, $f_{23}{}^1 = 1$, $f_{31}{}^2 = 1$). Therefore, the 6D Lie algebra is isomorphic to ISO(3) for any value of $\alpha$ as discussed in [28]. However, since the matrix (A.22) is not an element of O(3, 3), the redefined generators $T_A'$ do not have the canonical bilinear form: $\langle T_A', T_B' \rangle \neq \eta_{AB}$. According to [43], the only Drinfel'd double whose Lie algebra is isomorphic to ISO(3) is DD5. We have tried to find an O(3, 3) transformation which maps the algebra (A.20) to the Lie algebra of $(\mathbf{9}|\mathbf{1})$ but we could not find such an O(3, 3). We thus conclude that orbit 4 is related to a Drinfel'd double only when $\alpha = 0$.

**Orbit 5**

Orbit 5 contains the non-vanishing fluxes

$$H_{123} = c, \quad f_{12}{}^3 = s, \quad f_1{}^{23} = c, \quad f_2{}^{13} = c. \tag{A.23}$$

Here we consider two cases.

1. $\underline{\alpha = 0}$    In this case, performing an O(3, 3) transformation with

$$C_A{}^B = \begin{pmatrix} 0 & 0 & 0 & 0 & 0 & -\frac{1}{c} \\ 0 & 0 & 0 & -\frac{1}{\sqrt{2}} & \frac{1}{\sqrt{2}} & 0 \\ \frac{1}{\sqrt{2}} & \frac{1}{\sqrt{2}} & 0 & 0 & 0 & 0 \\ 0 & 0 & -c & 0 & 0 & 0 \\ -\frac{1}{\sqrt{2}} & \frac{1}{\sqrt{2}} & 0 & 0 & 0 & 0 \\ 0 & 0 & 0 & \frac{1}{\sqrt{2}} & \frac{1}{\sqrt{2}} & 0 \end{pmatrix}, \tag{A.24}$$

we get a Drinfel'd double with

$$f_{13}{}^2 = -t, \qquad f_{12}{}^2 = -1, \qquad f_{13}{}^3 = -1, \qquad f_3{}^{12} = c^2. \tag{A.25}$$

This is a Manin triple $(\mathbf{5}|\mathbf{2.ii})$, which is in the orbit

$$\text{DD6:} \qquad (\mathbf{8}|\mathbf{1}) \cong (\mathbf{8}|\mathbf{5.iii}) \cong (\mathbf{7_0}|\mathbf{5.i}) \cong (\mathbf{6_0}|\mathbf{5.i}) \cong (\mathbf{5}|\mathbf{2.ii}). \tag{A.26}$$

2. $\underline{\alpha \neq 0}$    In this case, performing an $O(3,3)$ transformation with

$$C_A{}^B = \begin{pmatrix} 0 & 0 & 0 & 0 & 0 & -\frac{1}{c} \\ -\frac{1}{\sqrt{2}t} & -\frac{1}{\sqrt{2}t} & 0 & 0 & 0 & 0 \\ 0 & 0 & 0 & -\frac{1}{\sqrt{2}} & \frac{1}{\sqrt{2}} & 0 \\ 0 & 0 & -c & 0 & 0 & 0 \\ 0 & 0 & 0 & -\frac{t}{\sqrt{2}} & -\frac{t}{\sqrt{2}} & 0 \\ -\frac{1}{\sqrt{2}} & \frac{1}{\sqrt{2}} & 0 & 0 & 0 & 0 \end{pmatrix}, \tag{A.27}$$

we get a Manin triple $(\mathbf{4}|\mathbf{2.iii}|b)$ with

$$f_{12}{}^2 = -1, \qquad f_{12}{}^3 = 1, \qquad f_{13}{}^3 = -1, \qquad f_2{}^{13} = -b \quad \left(b \equiv \frac{c^2}{t}\right). \tag{A.28}$$

This corresponds to the Drinfel'd double

$$\text{DD7:} \qquad (\mathbf{7_0}|\mathbf{4}|b) \cong (\mathbf{4}|\mathbf{2.iii}|b) \cong (\mathbf{6_0}|\mathbf{4.i}|-b) \qquad (b \neq 0). \tag{A.29}$$

According to [28], the Lie algebras of both Drinfel'd doubles are isomorphic to $CSO(2,1,1)$.

**Orbit 6**

Orbit 6 contains the non-vanishing fluxes

$$H_{123} = c, \quad f_{13}{}^2 = -s, \quad f_{12}{}^3 = s, \quad f_{12}{}^2 = f_{13}{}^3 = -\xi_0, \quad f_1{}^{23} = c, \quad Z_1 = -\xi_0. \tag{A.30}$$

This and the subsequent orbit contain non-vanishing $Z_A$ and the structure constants $X_{AB}{}^C$ have the symmetric part: $X_{(AB)}{}^C \neq 0$. If $\alpha = 0$, the flux configuration coincides with that of orbit 8 with $\alpha = 0$, which can be mapped to a $DD^+$ (which reduces to DD15 when $\xi_0 = 0$). When $\alpha \neq 0$, we fail to find an $O(3,3)$ transformation which maps this fluxes into any $DD^+$.[7]

To be a little more specific, let us consider the case $\xi_0 = 0$. By considering the eigenvalues of the Killing form, the only possible Drinfel'd doubles that may be related to orbit 6 are DD14–DD17. However, we could not find any $O(3,3)$ transformation which maps the flux configuration (A.30) with $\xi_0 = 0$ to any of these Drinfel'd double.[8] As we see below, DD14–DD17 rather correspond to orbit 7 or 8. We thus conclude that orbit 6 with $\xi_0 = 0$ can be related to a Drinfel'd double only when $\alpha = 0$.

---

[7]We considered a general $O(3,3)$ matrix $C = \begin{pmatrix} \alpha & 0 \\ 0 & (a^{-1})^t \end{pmatrix}\begin{pmatrix} 1 & \beta \\ 0 & 1 \end{pmatrix}\begin{pmatrix} 1 & 0 \\ \gamma & 1 \end{pmatrix}$ with $\det \alpha \neq 0$, $\beta^t = -\beta$, and $\gamma^t = -\gamma$, and tried to remove the flux components $F_{abc} = F^{abc} = 0$, but we could not find a real solution.

[8]A non-trivial solution we found is an $O(3,3;\mathbb{C})$ matrix

$$C_A{}^B = \begin{pmatrix} \frac{1}{c} & 0 & 0 & 0 & 0 & 0 \\ 0 & 0 & 0 & 0 & \frac{1}{2} & -\frac{i}{2} \\ 0 & \frac{i}{2} & \frac{1}{2} & 0 & 0 & 0 \\ 0 & 0 & 0 & c & 0 & 0 \\ 0 & 1 & i & 0 & 0 & 0 \\ 0 & 0 & 0 & 0 & -i & 1 \end{pmatrix},$$

which gives a Manin triple $(\mathbf{7_a}|\mathbf{1})$ with $\mathbf{a} = -i\,t$ pure imaginary: $f_{12}{}^3 = 1$, $f_{13}{}^2 = -1$, $f_{12}{}^2 = f_{13}{}^3 = -\mathbf{a}$.

**Orbit 7**

Orbit 7 contains the non-vanishing fluxes

$$H_{123} = c, \quad f_{13}{}^2 = -s, \quad f_{12}{}^3 = -s, \quad f_{12}{}^2 = f_{13}{}^3 = -\xi_0, \quad f_1{}^{23} = c, \quad Z_1 = -\xi_0. \quad (A.31)$$

If we perform an O(3,3) transformation with

$$C_A{}^B = \begin{pmatrix} \frac{1}{c} & 0 & 0 & 0 & 0 & 0 \\ 0 & 0 & 0 & 0 & \frac{1}{\sqrt{2}} & -\frac{1}{\sqrt{2}} \\ 0 & \frac{1}{\sqrt{2}} & \frac{1}{\sqrt{2}} & 0 & 0 & 0 \\ 0 & 0 & 0 & c & 0 & 0 \\ 0 & \frac{1}{\sqrt{2}} & -\frac{1}{\sqrt{2}} & 0 & 0 & 0 \\ 0 & 0 & 0 & 0 & \frac{1}{\sqrt{2}} & \frac{1}{\sqrt{2}} \end{pmatrix}, \quad (A.32)$$

we get a DD$^+$ with

$$f_{12}{}^3 = 1, \qquad f_{13}{}^2 = -1, \qquad f_{12}{}^2 = f_{13}{}^3 = -t - \frac{\xi_0}{c}, \qquad Z_1 = -\frac{\xi_0}{c}. \quad (A.33)$$

If we consider $\xi_0 = 0$, the DD$^+$ reduces to a Drinfel'd double, which can be classified as follows depending on the value of $\alpha$.

1. $\underline{0 < \alpha < \frac{\pi}{4}}$ This algebra is the Manin triple $(\mathbf{7_a}|\mathbf{1})$ with $\mathbf{a} = t$, which corresponds to

$$\text{DD14:} \qquad (\mathbf{7_a}|\mathbf{1}) \cong (\mathbf{7_a}|\mathbf{2.i}) \cong (\mathbf{7_a}|\mathbf{2.ii}) \quad (0 < \mathbf{a} < 1). \quad (A.34)$$

   The Lie algebra of this Drinfel'd double is CSO(2,0,2) [28].

2. $\underline{-\frac{\pi}{4} < \alpha < 0}$ In this case, the algebra can be mapped to the previous one through $T_1 \to -T_1$ and $T_2 \leftrightarrow T_3$.

3. $\underline{\alpha = \frac{\pi}{4}}$ In this case, the Killing form becomes the zero matrix and the Drinfel'd double has another name,

$$\text{DD18:} \qquad (\mathbf{7_1}|\mathbf{1}) \cong (\mathbf{7_1}|\mathbf{2.i}) \cong (\mathbf{7_1}|\mathbf{2.ii}). \quad (A.35)$$

   In [28], the Lie algebra of this Drinfel'd double is denoted as $\mathfrak{g}_0$.

4. $\underline{\alpha = 0}$ In this case, the embedding tensor is the same as that of orbit 8 with $\alpha = 0$, which is studied below.

**Orbit 8**

Orbit 8 contains the non-vanishing fluxes

$$H_{123} = c, \quad f_{12}{}^3 = s, \quad f_{12}{}^2 = f_{13}{}^3 = -\xi_0, \quad f_1{}^{23} = c, \quad Z_1 = -\xi_0. \quad (A.36)$$

Performing an O(3,3) transformation with

$$C_A{}^B = \begin{pmatrix} 0 & 0 & 0 & 0 & 1 & 0 \\ 0 & 0 & 1 & 0 & 0 & 0 \\ \frac{1}{c} & 0 & 0 & 0 & 0 & 0 \\ 0 & 1 & 0 & 0 & 0 & 0 \\ 0 & 0 & 0 & 0 & 0 & 1 \\ 0 & 0 & 0 & c & 0 & 0 \end{pmatrix}, \quad (A.37)$$

we get a DD$^+$ with

$$f_{23}{}^1 = 1, \quad f_{13}{}^2 = -1, \quad f_{13}{}^1 = f_{23}{}^2 = \frac{\xi_0}{c}, \quad f_3{}^{12} = t, \quad Z_3 = -\frac{\xi_0}{c}. \quad (A.38)$$

Again, let us consider the reduction to the Drinfel'd double $\xi_0 = 0$, which can be decomposed into the following three cases.

1. $\underline{\alpha = 0}$    In this case, we have

$$\text{DD15:} \qquad (\mathbf{7_0}|\mathbf{1}). \tag{A.39}$$

2. $\underline{0 < \alpha < \frac{\pi}{2}}$    Through a rescaling of $T_1$ and $T_2$, we obtain

$$\text{DD16:} \qquad (\mathbf{7_0}|\mathbf{2.i}). \tag{A.40}$$

3. $\underline{-\frac{\pi}{2} < \alpha < 0}$    Through a rescaling of $T_1$ and $T_2$, we have

$$\text{DD17:} \qquad (\mathbf{7_0}|\mathbf{2.ii}). \tag{A.41}$$

In any of these cases, the Lie algebra of the Drinfel'd double is isomorphic to $\mathfrak{h}_1$ of [28].

**Orbit 9**

Orbit 9 contains the non-vanishing fluxes

$$H_{123} = c, \quad f_{13}{}^2 = -s, \quad f_{12}{}^3 = s, \quad f_{12}{}^2 = f_{13}{}^3 = -\xi_0, \quad f_1{}^{23} = -c, \quad Z_1 = -\xi_0. \tag{A.42}$$

This can be mapped to the following three DD$^+$s.

1. $\underline{\alpha = 0}$    In this case, the embedding tensor is the same as that of orbit 11 with $\alpha = 0$.

2. $\underline{\alpha \neq 0}$    Performing an O(3,3) transformation with

$$C_A{}^B = \begin{pmatrix} -\frac{1}{s} & 0 & 0 & 0 & 0 & 0 \\ 0 & \frac{1}{\sqrt{t}} & 0 & 0 & 0 & \frac{1}{\sqrt{t}} \\ 0 & 0 & -\frac{1}{\sqrt{t}} & 0 & \frac{1}{\sqrt{t}} & 0 \\ 0 & 0 & 0 & -s & 0 & 0 \\ 0 & 0 & 0 & 0 & \sqrt{t} & 0 \\ 0 & 0 & 0 & 0 & 0 & -\sqrt{t} \end{pmatrix}, \tag{A.43}$$

we get a DD$^+$ with

$$f_{12}{}^3 = 1, \quad f_{13}{}^2 = -1, \quad f_{12}{}^2 = f_{13}{}^3 = -\frac{1}{t} + \frac{\xi_0}{s}, \quad f_1{}^{23} = -1, \quad Z_1 = \frac{\xi_0}{s}. \tag{A.44}$$

Now, let us consider the case $\xi_0 = 0$. If $0 < \alpha < \frac{\pi}{4}$, this is a Manin triple $(\mathbf{7_a}|\mathbf{2.ii})$ with $\mathbf{a} \equiv \frac{1}{t}$, which corresponds to

$$\text{DD9:} \qquad (\mathbf{7_a}|\mathbf{1}) \cong (\mathbf{7_a}|\mathbf{2.i}) \cong (\mathbf{7_a}|\mathbf{2.ii}) \quad (\mathbf{a} > 1). \tag{A.45}$$

When $\alpha$ is negative, we can consider a redefinition, such as $T_1 \to -T_1$ and $T_2 \to -T_2$, which flips the sign of $t$, and the Drinfel'd double is always DD9.

3. $\underline{\alpha = \frac{\pi}{4}}$    This case is the same as the previous case by choosing $\alpha = \frac{\pi}{4}$.

   When $\xi_0 = 0$, the Drinfel'd double has another name

$$\text{DD18:} \qquad (\mathbf{7_1}|\mathbf{1}) \cong (\mathbf{7_1}|\mathbf{2.i}) \cong (\mathbf{7_1}|\mathbf{2.ii}), \tag{A.46}$$

because the number of the null eigenvalues of the Killing form is increased.

**Orbit 10**

Orbit 10 contains the non-vanishing fluxes

$$H_{123} = c, \quad f_{13}{}^2 = f_{12}{}^3 = -s, \quad f_{12}{}^2 = f_{13}{}^3 = -\xi_0, \quad f_1{}^{23} = -c, \quad Z_1 = -\xi_0. \tag{A.47}$$

Performing an O(3,3) transformation with

$$C_A{}^B = \begin{pmatrix} \frac{1}{s} & 0 & 0 & 0 & 0 & 0 \\ 0 & \frac{1}{\sqrt{2}} & 0 & 0 & 0 & -\frac{1}{\sqrt{2}} \\ 0 & 0 & \frac{1}{\sqrt{2}} & 0 & \frac{1}{\sqrt{2}} & 0 \\ 0 & 0 & 0 & s & 0 & 0 \\ 0 & 0 & -\frac{1}{\sqrt{2}} & 0 & \frac{1}{\sqrt{2}} & 0 \\ 0 & \frac{1}{\sqrt{2}} & 0 & 0 & 0 & \frac{1}{\sqrt{2}} \end{pmatrix}, \tag{A.48}$$

we get a DD$^+$ with

$$f_{12}{}^3 = f_{13}{}^2 = -1, \quad f_{12}{}^2 = f_{13}{}^3 = -\frac{1}{t} - \frac{\xi_0}{s}, \quad Z_1 = -\frac{\xi_0}{s}. \tag{A.49}$$

If we consider the case $\xi_0 = 0$, we obtain the following Drinfel'd doubles.

1. $\underline{\alpha = 0}$   Again, the embedding tensor is the same as that of orbit 11 with $\alpha = 0$.

2. $\underline{0 < \alpha < \frac{\pi}{4}}$   This is the Manin triple $(\mathbf{6_a}|\mathbf{1})$ with $\mathbf{a} \equiv \frac{1}{t}$, which corresponds to[9]

$$\text{DD10:} \qquad (\mathbf{6_a}|\mathbf{1}) \cong (\mathbf{6_a}|\mathbf{2}) \cong (\mathbf{6_a}|\mathbf{6_{1/a}.ii}) \cong (\mathbf{6_a}|\mathbf{6_{1/a}.iii}) \quad (\mathbf{a} > 1). \tag{A.50}$$

3. $\underline{-\frac{\pi}{4} < \alpha < 0}$   This can be mapped to the previous case through $T_1 \to -T_1$, $T_2 \to T_3$, and $T_3 \to -T_2$.

4. $\underline{\alpha = \frac{\pi}{4} \ (t = 1)}$

$$\text{DD13:} \qquad (\mathbf{3}|\mathbf{1}) \cong (\mathbf{3}|\mathbf{2}) \cong (\mathbf{3}|\mathbf{3.ii}) \cong (\mathbf{3}|\mathbf{3.iii}). \tag{A.51}$$

When $\alpha \neq 0$, the Lie algebras of the Drinfel'd doubles are isomorphic to CSO(1, 1, 2) [28].

**Orbit 11**

Orbit 11 contains the non-vanishing fluxes

$$H_{123} = c, \quad f_{12}{}^3 = s, \quad f_{12}{}^2 = f_{13}{}^3 = -\xi_0, \quad f_1{}^{23} = -c, \quad Z_1 = -\xi_0. \tag{A.52}$$

Under an O(3,3) transformation

$$C_A{}^B = \begin{pmatrix} 0 & 0 & 0 & 0 & 1 & 0 \\ 0 & 0 & -1 & 0 & 0 & 0 \\ -\frac{1}{c} & 0 & 0 & 0 & 0 & 0 \\ 0 & 1 & 0 & 0 & 0 & 0 \\ 0 & 0 & 0 & 0 & 0 & -1 \\ 0 & 0 & 0 & -c & 0 & 0 \end{pmatrix}, \tag{A.53}$$

---

[9]We note that the Manin triple $(\mathbf{6_a}|\mathbf{1})$ can be mapped to $(\mathbf{6_{1/a}}|\mathbf{1})$, for example, through

$$C_A{}^B = \begin{pmatrix} -\frac{1}{a} & 0 & 0 & 0 & 0 & 0 \\ 0 & \frac{1}{2} & -\frac{1}{2} & 0 & \frac{1}{2} & \frac{1}{2} \\ 0 & -\frac{1}{2} & \frac{1}{2} & 0 & \frac{1}{2} & \frac{1}{2} \\ 0 & 0 & 0 & -a & 0 & 0 \\ 0 & \frac{1}{2} & \frac{1}{2} & 0 & \frac{1}{2} & -\frac{1}{2} \\ 0 & \frac{1}{2} & \frac{1}{2} & 0 & -\frac{1}{2} & \frac{1}{2} \end{pmatrix}.$$

Thus the parameter $\mathbf{a}$ of $(\mathbf{6_a}|\mathbf{1})$ can be restricted to $\mathbf{a} > 1$.

we get a DD$^+$ with

$$f_{13}{}^2 = 1\,, \quad f_{23}{}^1 = 1\,, \quad f_{13}{}^1 = f_{23}{}^2 = -\frac{\xi_0}{c}\,, \quad f_3{}^{12} = t\,, \quad Z_3 = \frac{\xi_0}{c}\,. \qquad \text{(A.54)}$$

For the case of the Drinfel'd double ($\xi_0 = 0$), this can be classified into two Drinfel'd doubles.

1. $\underline{\alpha = 0}$  The Manin triple is $(\mathbf{6_0}|\mathbf{1})$, which is contained in

$$\text{DD11:} \qquad (\mathbf{6_0}|\mathbf{1}) \cong (\mathbf{6_0}|\mathbf{5.ii}) \cong (\mathbf{5}|\mathbf{1}) \cong (\mathbf{5}|\mathbf{2.i})\,. \qquad \text{(A.55)}$$

2. $\underline{\alpha \neq 0}$  Performing a rescaling of $T_1$ and $T_2$, we get the Manin triple $(\mathbf{6_0}|\mathbf{2})$, which is contained in

$$\text{DD12:} \qquad (\mathbf{6_0}|\mathbf{2}) \cong (\mathbf{6_0}|\mathbf{4.ii}) \cong (\mathbf{4}|\mathbf{1}) \cong (\mathbf{4}|\mathbf{2.i}) \cong (\mathbf{4}|\mathbf{2.ii})\,. \qquad \text{(A.56)}$$

The Lie algebras of both Drinfel'd doubles are called $\mathfrak{h}_2$ [28].

**Orbit 12**

Orbit 12 contains the non-vanishing fluxes

$$H_{123} = c\,, \quad f_{12}{}^3 = s\,, \quad f_{12}{}^2 = f_{13}{}^3 = -\xi_0\,, \quad Z_1 = -\xi_0\,. \qquad \text{(A.57)}$$

Under an O(3,3) transformation with

$$C_A{}^B = \begin{pmatrix} 0 & 0 & 0 & 0 & c & 0 \\ 0 & 0 & 1 & 0 & 0 & 0 \\ 1 & 0 & 0 & 0 & 0 & 0 \\ 0 & \frac{1}{c} & 0 & 0 & 0 & 0 \\ 0 & 0 & 0 & 0 & 0 & 1 \\ 0 & 0 & 0 & 1 & 0 & 0 \end{pmatrix}\,, \qquad \text{(A.58)}$$

we get a DD$^+$ with

$$f_{23}{}^1 = 1\,, \quad f_{13}{}^1 = f_{23}{}^2 = \xi_0\,, \quad f_3{}^{12} = t\,, \quad Z_3 = -\xi_0\,. \qquad \text{(A.59)}$$

When $\xi_0 = 0$, the Drinfel'd double can be classified as follows.

1. $\underline{\alpha = 0}$

$$\text{DD19:} \qquad (\mathbf{2}|\mathbf{1})\,. \qquad \text{(A.60)}$$

2. $\underline{0 < \alpha \leq \frac{\pi}{4}}$  Through a rescaling of $T_1$ and $T_2$, we get

$$\text{DD20:} \qquad (\mathbf{2}|\mathbf{2.i})\,. \qquad \text{(A.61)}$$

3. $\underline{-\frac{\pi}{4} < \alpha < 0}$  Through a rescaling of $T_1$ and $T_2$, we get

$$\text{DD21:} \qquad (\mathbf{2}|\mathbf{2.ii})\,. \qquad \text{(A.62)}$$

The Lie algebras of these Drinfel'd double are isomorphic to CSO(1,0,3) [28].

Table A.1: Correspondence between 13 orbits of [28] and 22 Drinfel'd doubles classified in [43]. DD1, 2, 3, 4, and 8 contain a parameter $b$ whose range is shown in round brackets. Only a specific value is realized if we construct the Drinfel'd double from the flux algebras of [28].

| DD | DD1 $b=1$ $(b>0)$ | DD2 $b=1$ $(b>0)$ | DD3 $b=\frac{a}{1+a^2}$ $(b\neq 0)$ | DD4 $b=\frac{a}{1+a^2}$ $(b\neq 0)$ | DD5 | DD6 | DD7 | DD8 $b=\frac{1}{2}$ $(b\neq 0)$ | DD9 | DD10 | DD11 |
|---|---|---|---|---|---|---|---|---|---|---|---|
| Orbit | 2 | 3 | 2 | 3 | 4 | 5 | 5 | 3 | 9 | 10 | 9,10,11 |

| DD | DD12 | DD13 | DD14 | DD15 | DD16 | DD17 | DD18 | DD19 | DD20 | DD21 | DD22 |
|---|---|---|---|---|---|---|---|---|---|---|---|
| Orbit | 11 | 10 | 7 | 6,7,8 | 8 | 8 | 7,9 | 12 | 12 | 12 | 13 |

## Orbit 13

Orbit 13 contains only $Z_1 = -\xi_0$. Without any redefinition of generators, this corresponds to the Jacobi–Lie bialgebra $((I, -2\check{X}^1), (I, 0))$. In the case $\xi_0$, we get an Abelian double, called DD22.

## Summary

We can summarize the result as in Table A.1. We found that all of the 22 Drinfel'd doubles are reproduced from the 13 orbits of the embedding tensors in 7D supergravity by choosing $\xi_0 = 0$. The parameter $b$ contained in the Lie algebra of several Drinfel'd doubles takes the specific value listed in Table A.1. This is natural because the Lie algebra of the Drinfel'd double does not depend on the parameter $b$, and the classification made in [28] is the classification of the Lie algebra without considering the bilinear form $\langle \cdot, \cdot \rangle$. In order to realize a Drinfel'd double with a different value of $b$, we need to change the bilinear form $\langle \cdot, \cdot \rangle$, which corresponds to performing a non-O(3,3) redefinition of generators. For DD1 and DD2, we can perform a non-O(3,3) redefinition of generators, $T'_a = T_a$ and $T'^a = b\,T^a$, to convert the value of the parameter $b$ from the special value of 1 to the general value of $b$. Similarly, for DD3, DD4, and DD8, the redefinition $T'_a = T_a$ and $T'^a = \frac{b}{cs}\,T^a$ change the value of $b$. This way, all of the Drinfel'd doubles classified in [43] can be reproduced from the orbits classified in [28].

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
