# Peer review of "Jacobi-Lie T-plurality"

_SciPost Physics, doi:SciPost Phys. 11, 038 (2021)_

## Round 1 · Referee Report · Anonymous (Referee 1) · 2021-5-25

Report

The authors introduce the notion of the "extended Drinfeld algebra" (EDA) which differ from the standard Drinfeld double Lie algebra by admiting
a non-vanishing symmetric part of the structure constants and by violating the invariance of the bilinear form by admitting its scaling. The authors then show that their generalization permit to construct generalized frames (GE) the generalized Lie derivatives of which are the constant linear combinations of GEs with the constants being just the structure constants of the EDA. The authors then generalize the Poisson-Lie T-plurality to the Jacobi-Lie T-plurality not at the level of the sigma-model but at the level of the symmetries of the double field theory. I appreciate the result still I am tempted to believe that with a little bit of more effort the authors could probable obtain the Jacobi-Lie canonical equivalence of the sigma-models living on the corresponding backgrounds. I buy the arguments of the authors why it is not easy to figure out some generalization of the E-model which would establish this sigma-model equivalence at a general level, but why not to study some of the low dimensional examples of the Jacobi-Lie T-plurality just by "hand" by trying to construct the needed canonical transformation relating the DFT Jacobi-Lie equivalent backgrounds? Maybe this effort could lead to a general notion of the sigma model Jacobi-Lie T-duality?

Anyway, the article is correct, it is well written, it gives new results which fit well in a broader and rapidly developing context so I recommend it for publication. Just some more precisions could be perhaps added, for example, it is not quite explained in the paper why the rotation (3.41) should be considered as the generalized Yang-Baxter deformation (neither there is a reference what the YB deformation is so that the reader could figure it out by himself).
  • validity: high
  • significance: high
  • originality: high
  • clarity: good
  • formatting: excellent
  • grammar: good

Author:  Jose J. Fernandez-Melgarejo  on 2021-07-21  [id 1599]

(in reply to Report 1 on 2021-05-25)

Thank you for positive comments and useful suggestions. For the Yang-Baxter deformation, we have added a short explanation, around Eq.(3.47), why the O(4,4) rotation is called the generalized Yang-Baxter deformation. Regarding a concrete example of the Jacobi-Lie T-plurality at the level of sigma model, we could not find a useful example. As we have explained on page 29 of the revised manuscript, in the Jacobi-Lie case, we cannot regard the currents $\mathcal{J}_A$ as the phase-space variables. Then it is not clear for us how to describe the Jacobi-Lie T-plurality as a canonical transformation. We hope to clarify this point in future research.

---

## Round 1 · Referee Report · Anonymous (Referee 2) · 2021-6-8

Strengths

1 - Elegant extension of Drinfel'd double to a larger algebraic structure.
2 - Gives a new solution-generating mechanism.
3 - Solves some outstanding issues in the definition/application of Jacobi-Lie bialgebras.
4 - Several examples are given of the new method.
5 - Well written.
6 - Extension of interesting research in this direction, which has seen many developments in recent years and is likely to lead to significant new insight in future.

Weaknesses

1 - Little introduction to double field theory, doubled coordinates, Jacobi-Lie bialgebras and Jacobi-Lie structures. This makes it difficult to follow for anyone but a very tiny number of researchers.
2 - Some arguments are murky. In particular:
a) the need for $Z^a$ = 0
b) why the sigma-model EoMs cannot be shown to be invariant under the transformations considered
c) Are all 7-d half-max gaugings related by an O(d,d) transformation to a $DD^+$ algebra? This seems surprising and contradictory statements seem to be made on page 11.

Report

This is a very good paper, with overall a nice presentation, and useful results extending Drinfel'd Doubles and their associated Poisson-Lie T-duality to a larger class of $DD^+$ algebras and associated "Jacobi-Lie T-Plurality". This forms a nice new supergravity solution-generating mechanism, with several examples given. The authors do a very good job at relating their paper to other investigations of Jacobi-Lie bialgebras in the literature. In particular, the authors here solve a mystery observed in their reference [17] with regards to Jacobi-Lie bialgebras. This is really very nice.

My main issues are already described in the "Weaknesses" section of this report. Let me emphasise those which I would most like to see the authors address.

1) It would be very useful to have a short section reviewing Jacobi-Lie bialgebras, Jacobi-Lie structures and their relation to contact geometry, esp. Reeb vectors. This is very specialised knowledge which all but a tiny handful of readers would be lacking. This material could be included as a recap section in the main part of the paper or as an appendix, and I feel it would really round the paper off nicely.

2) Some of the important arguments made are not terribly clear. In particular, on page 3 in the introduction, the authors state that they must take $Z^a = 0$ to "exhibit the O(d,d) covariance of the DFT equations of motion". What exactly does this mean? I could not find a place where this is explained in more detailed in the later parts of the section. It would be nice to have this spelled out explicitly in the paper. In particular, from (2.29) it seems like $Z^a \neq 0$ is related to having a linear dependence on a dual coordinate. This seems reminiscent to generalised supergravity. Is there a connection?

3) What is the reason that the sigma-model EoMs cannot be shown to be invariant under the Jacobi-Lie T-Plurality? Is there an issue with having a transformation acting within O(d,d) $\times \mathbb{R}^+$ and this somehow shifting the dilaton in a difficult way?

4) On page 11, there seem to be contradictory statements made about whether all 7-d half-max gaugings are related by an O(d,d) transformation to a $DD^+$ algebra. This almost certainly should not be the case since I would have thought that a generic 7-d half-max gauging does not have two isotropic subalgebras. Indeed, later on page 11 the authors themselves state that "Despite some of the embedding tensor configurations may not be mapped to any $DD^+$". The authors should be more precise here.

Apart from this, there are some other small things that I believe would improve the paper:

a) State that in (1.1), the $f_{ab}{}^c$ are the structure constants of a Lie algebra. b) Explain what the $\tilde{\partial}^m$ derivative is in (2.29) -- at least heuristically for those who do not know DFT. c) The notation at the bottom of page 12, top of page 13, in particular, ${\cal E}_{mn}$ vs ${\cal E}_{ab}$ is a bit confusing. It is very tempting to assume that

$$ {\cal E}{mn} = r^a_m r^b_n {\cal E} $$
but this seems to not be the case because there is also a rescaling by $e^{-2\Delta}$. Perhaps the authors could use a different symbol instead of ${\cal E}_{mn}$ for the rescaled quantity? At the very least, the authors might want to emphasise this subtlety.

Another, less trivial question is the following, but this may be asking too much for this paper: Can the authors more generally prove that the algebra with $Z^a \neq 0$ cannot come from the standard solution of the section condition? Their current argument seems to hugely rely on the form of the Ansatz for $E_A{}^M$ in (2.25). Can they prove something about completely general $E_A{}^M$?

Requested changes

Here is a minimum list of requested changes. 1) It would be very useful to have a short section reviewing Jacobi-Lie bialgebras, Jacobi-Lie structures and their relation to contact geometry, esp. Reeb vectors. This is very specialised knowledge which all but a tiny handful of readers would be lacking. This material could be included as a recap section in the main part of the paper or as an appendix, and I feel it would really round the paper off nicely.

2) Some of the important arguments made are not terribly clear. In particular, on page 3 in the introduction, the authors state that they must take $Z^a = 0$ to "exhibit the O(d,d) covariance of the DFT equations of motion". What exactly does this mean? I could not find a place where this is explained in more detailed in the later parts of the section. It would be nice to have this spelled out explicitly in the paper. In particular, from (2.29) it seems like $Z^a \neq 0$ is related to having a linear dependence on a dual coordinate. This seems reminiscent to generalised supergravity. Is there a connection?

3) What is the reason that the sigma-model EoMs cannot be shown to be invariant under the Jacobi-Lie T-Plurality? Is there an issue with having a transformation acting within O(d,d) $\times \mathbb{R}^+$ and this somehow shifting the dilaton in a difficult way?

4) On page 11, there seem to be contradictory statements made about whether all 7-d half-max gaugings are related by an O(d,d) transformation to a $DD^+$ algebra. This almost certainly should not be the case since I would have thought that a generic 7-d half-max gauging does not have two isotropic subalgebras. Indeed, later on page 11 the authors themselves state that "Despite some of the embedding tensor configurations may not be mapped to any $DD^+$". The authors should be more precise here.

The authors may also want to address the other questions I listed in the report, labelled a) -- c) and the final question I give in the report.

  • validity: high
  • significance: high
  • originality: good
  • clarity: good
  • formatting: excellent
  • grammar: excellent

Author:  Jose J. Fernandez-Melgarejo  on 2021-07-21  [id 1598]

(in reply to Report 2 on 2021-06-08)

1) Thank you for the useful suggestions. In the revised manuscript, we have added the definition of the Jacobi bracket Eq.(2.37), and explained that Eq.(2.32) is equivalent to the Jacobi identities for the Jacobi bracket. We have also explained why the pair of the bi-vector $\pi$ and a vector field $E$ is called the Jacobi-Lie structure. In the original manuscript, we used a terminology Reeb vector, but this was not appropriate and has been removed in the revised manuscript.

2) We would like to appreciate your useful comments. Originally we considered that $Z^a = 0$ is necessary, but it was our mistake. In the revised manuscript, we have extended the discussion by removing the assumption $Z^a = 0$, and provided an example of the Jacobi-Lie T-plurality with non-vanishing $Z^a$ (section 3.3.3). Unlike the (generalized) supergravity (where only the dilaton can have dual-coordinate dependence), the metric and the B-field have an overall factor that has a linear dependence on the dual coordinates. However, the section condition of DFT is not broken and this is a solution of DFT. It may be also possible to understand this background as a solution of a certain deformed supergravity.

As discussed on page 17, there are some difficulties if both $Z^a$ and $f_b{}^{ba}$ do not vanish and they are linearly independent. We have provided a concrete example of such a problematic case in section 3.4.1.

3) Concerning the covariance in the sigma-model, we have slightly clarified the explanation of section 4. In the case of the $\mathcal{E}$-model (i.e., string theory on a Poisson-Lie symmetric target space), the equations of motion are described by the Hamiltonian (4.1) and the current algebra (4.2). After performing the Poisson-Lie T-duality, the Hamiltonian becomes (4.7) and the current algebra becomes (4.8). In the both duality frame, the only dynamical variables are the currents $j_A(\sigma)$ or $j'_A(\sigma)$. In the literature, the currents are interpreted as the phase-space variables and then the current algebras correspond to the canonical commutation relation. The Hamiltonian is simply a bilinear form of the phase-space variables. Under the identification $j'_A = C_A{}^B j_B$, the Hamiltonian and the canonical commutation relation are not changed under the Poisson-Lie T-duality. Consequently, the Poisson-Lie T-duality has been understood as a canonical transformation.

In the case of the Jacobi-Lie T-duality, the Hamiltonian has the same form (4.12), but the current algebra (4.13) additionally contains $\omega(x(\sigma))$. Unlike the case of the Poisson-Lie T-duality, the embedding functions $x(\sigma)$ of the string explicitly appear in the algebra, and we cannot regard $j_A(\sigma)$ as the fundamental variables: $j_A(\sigma)$ corresponds to complicated functions of $p_m$ and $x^m$ as described in (4.5). Then the Hamiltonian (4.12) becomes a highly non-linear function. Under such a situation, we have no idea how to understand the O(D,D) rotation as a canonical transformation.

4) Thank you for pointing out the unclear statements. As was mentioned in our original manuscript, it is not be possible to relate all 7D half-max gaugings to a DD$^+$. In order to clarify which gaugings can be related to DD$^+$, we have added Appendix A. Among the 13 orbits classified in the literature, almost all orbits (except 1, 4, 6) can be related to DD$^+$. Orbits 4 and 6 can be related to some DD$^+$ if the parameter $\alpha=0$. Orbit 1 cannot be related to any DD$^+$. We have also modified section 2.5 accordingly.

Regarding other questions a)--c):

Thank you for the suggestions. We have modified the manuscript as explained below.

a) We have added some explanation below Eq. (1.1). b) We have added the definitions of $\tilde{\partial}^m$ below Eq.(2.27). c) We have removed the misleading notation $\mathcal{E}_{ab}$.

Reply to the question "on the algebra with $Z^a \neq 0$":

In the revised manuscript, the restriction $Z^a = 0$ has been removed, so we believe this question has been resolved. When $Z^a \neq 0$, the frame fields (2.28) (of the revised manuscript) work nicely and the section condition is not broken.

---

## Round 2 · Referee Report · Anonymous (Referee 1) · 2021-7-21

Report

I am satisfied with the revised version of the paper and I recommend it for publication

---

## Round 2 · Referee Report · Anonymous (Referee 2) · 2021-8-7

Report

The authors have made the changes I requested in my previous report and I am happy to now recommend publication.

---

## Editorial Decision

published